# A metabolic modeling-based framework for predicting trophic dependencies in native rhizobiomes of crop plants

Alon Avraham Ginatt[1,2]*, Maria Berihu[1], Einam Castel[1], Shlomit Medina[1], Gon Carmi[3], Adi Faigenboim-Doron[4], Itai Sharon[5,6], Ofir Tal[7], Samir Droby[8], Tracey Somera[9], Mark Mazzola[10], Hanan Eizenberg[1], Shiri Freilich[1]

[1]Department of Natural Resources, Newe Ya'ar Research Center, Agricultural Research Organization (Volcani Institute), Ramat Ishay, Israel; [2]Department of Plant Pathology and Microbiology, The Robert H. Smith Faculty of Agriculture, Food and Environment, The Hebrew University of Jerusalem, Rehovot, Israel; [3]Bioinformatics Unit, Newe Ya'ar Research Center, Agricultural Research Organization (Volcani Institute), Ramat Yishay, Israel; [4]Institute of Plant Sciences, Agricultural Research Organization (ARO), The Volcani Center, Beit Dagan, Israel; [5]Migal-Galilee Research Institute, Kiryat Shmona, Israel; [6]Faculty of Sciences and Technology, Tel-Hai Academic College, Qiryat Shemona, Israel; [7]Kinneret Limnological Laboratory, Israel Oceanographic and Limnological Research, Migdal, Israel; [8]Department of Postharvest Sciences, Agricultural Research Organization (ARO), The Volcani Center, Rishon LeZion, Israel; [9]United States Department of Agriculture-Agricultural Research Service Tree Fruits Research Lab, Wenatchee, United States; [10]Department of Plant Pathology, Stellenbosch University, Stellenbosch, South Africa

*For correspondence:
Alon.ginatt@protonmail.com

Competing interest: The authors declare that no competing interests exist.

## eLife assessment

By developing a framework to integrate metagenomic and metabolomic data with genome-scale metabolic models, this study establishes a toolkit to investigate trophic interactions between microbiota members in situ. The authors apply this method to the native rhizosphere bacterial communities of apple rootstocks, producing **solid** evidence and numerous detailed hypotheses on specific trophic exchanges and resource dependencies. The framework represents a **valuable** method to disentangle features of microbial interaction networks and will be of interest to microbiome scientists as well as plant and computational biologists.

**Abstract** The exchange of metabolites (i.e., metabolic interactions) between bacteria in the rhizosphere determines various plant-associated functions. Systematically understanding the metabolic interactions in the rhizosphere, as well as in other types of microbial communities, would open the door to the optimization of specific predefined functions of interest, and therefore to the harnessing of the functionality of various types of microbiomes. However, mechanistic knowledge regarding the gathering and interpretation of these interactions is limited. Here, we present a framework utilizing genomics and constraint-based modeling approaches, aiming to interpret the hierarchical trophic interactions in the soil environment. 243 genome scale metabolic models of bacteria associated with a specific disease-suppressive vs disease-conducive apple rhizospheres were drafted based on genome-resolved metagenomes, comprising an in silico native microbial community. Iteratively simulating microbial community members' growth in a metabolomics-based apple root-like environment produced novel data on potential trophic successions, used to form a network of

communal trophic dependencies. Network-based analyses have characterized interactions associated with beneficial vs non-beneficial microbiome functioning, pinpointing specific compounds and microbial species as potential disease supporting and suppressing agents. This framework provides a means for capturing trophic interactions and formulating a range of testable hypotheses regarding the metabolic capabilities of microbial communities within their natural environment. Essentially, it can be applied to different environments and biological landscapes, elucidating the conditions for the targeted manipulation of various microbiomes, and the execution of countless predefined functions.

## Introduction

The rhizosphere serves as a hotspot for a diversity of interactions spanning from the secretion of organic compounds by plant roots to their uptake by the adjacent soil microbial community (*Zhalnina et al., 2018*; *Gomariz et al., 2015*). These interactions form a complex network of metabolic exchanges whose structure and function has a considerable impact on plant health (*Singh et al., 2004*). Targeted secretion of exudates from plant roots' into the environment is fundamental to the recruitment of specific microbes and the assembly of a plant-selected community (i.e., the rhizobiome) (*Korenblum et al., 2020*; *Sasse et al., 2018*; *Venturi and Keel, 2016*). Each plant has a unique profile of exudates guiding the formation of a specialized rhizobiome that is adapted to support its mineral absorption (*Olanrewaju et al., 2017*; *Rawat et al., 2021*; *Compant et al., 2010*), secrete plant growth supporting compounds (*Finkel et al., 2020*; *Ghosh et al., 2020*; *Kudoyarova et al., 2019*), and provide protection against soil-borne microorganisms that are detrimental to its health (*Berendsen et al., 2012*; *Ngalimat et al., 2021*; *Mazzola and Freilich, 2017*). A comprehensive understanding of the dynamics within the rhizosphere, considering both plant–microbe (PM) and microbe–microbe interactions, can guide the targeted assembly and maintenance of plant-beneficial soil microbial systems (*Freilich et al., 2011*; *Embree et al., 2015*; *Tsoi et al., 2018*; *Zengler and Zaramela, 2018*). The development of such microbiome-based, plant-beneficial strategies presents ecologically sound alternatives to conventional, chemical-based solutions in supporting plant health and productivity (*Vessey, 2003*; *Toju et al., 2018*).

`Omics data in general and specifically metagenomics data analyses can potentially provide keys for unraveling the black box of PM and microbe–microbe interactions in complex ecosystems such as the soil (*Faust and Raes, 2012*; *Widder et al., 2016*; *Magnúsdóttir et al., 2017*; *San León and Nogales, 2022*). Sequence-based analyses are, however, typically limited in terms of functional interpretation of community dynamics (*Basile et al., 2020*). Constraint-based modeling (CBM) is an approach that allows for the simulation of bacterial-metabolic activity in a given environment based on the constraints imposed by the annotated microbial genomes (*Faust and Raes, 2012*; *Orth et al., 2010*). This approach has long been used for studying the physiology and growth of single cells, represented as genome scale metabolic models (GSMMs), under varying conditions (*Cuevas et al., 2016*; *Price et al., 2004*). Applying CBM over a GSMM can be used to assess the uptake and secretion of metabolites in the environment under study (*Orth et al., 2010*). Accordingly, when CBM is applied over multiple GSMMs, metabolite exchange profiles (secretion into and consumption from the environment) become interconnected. This sheds light on conditions supporting the growth of different bacterial groups within the community as well as their functional potential in the trophic network formed (*Freilich et al., 2011*; *Stolyar et al., 2007*).

Advancement of sequencing technologies alongside the development of automatic pipelines for GSMM construction (*Machado et al., 2018*; *Henry et al., 2010*) has promoted an increased use of CBM for the modeling of communities with growing complexity (*Faust and Raes, 2012*; *Widder et al., 2016*; *Magnúsdóttir et al., 2017*; *Basile et al., 2020*; *Zampieri et al., 2023*; *Heinken et al., 2023*). The relevance of CBM to the study of microbiomes has been demonstrated in a variety of ecosystems and recent works have shown that CBM-based predictions can guide the development of strategies for microbiome management (*Heinken et al., 2023*; *Faust, 2019*; *Xu et al., 2019*; *Ruan et al., 2022*). An accurate representation of microbial metabolic networks depends on the origin of the genomes analyzed. To date, most studies attempting to model the microbiome of specific ecosystems by GSMM represent native species using corresponding genome sequences from public depositories, a process which is usually referred to as 16S-based genome imputation (*Basile et al.,*

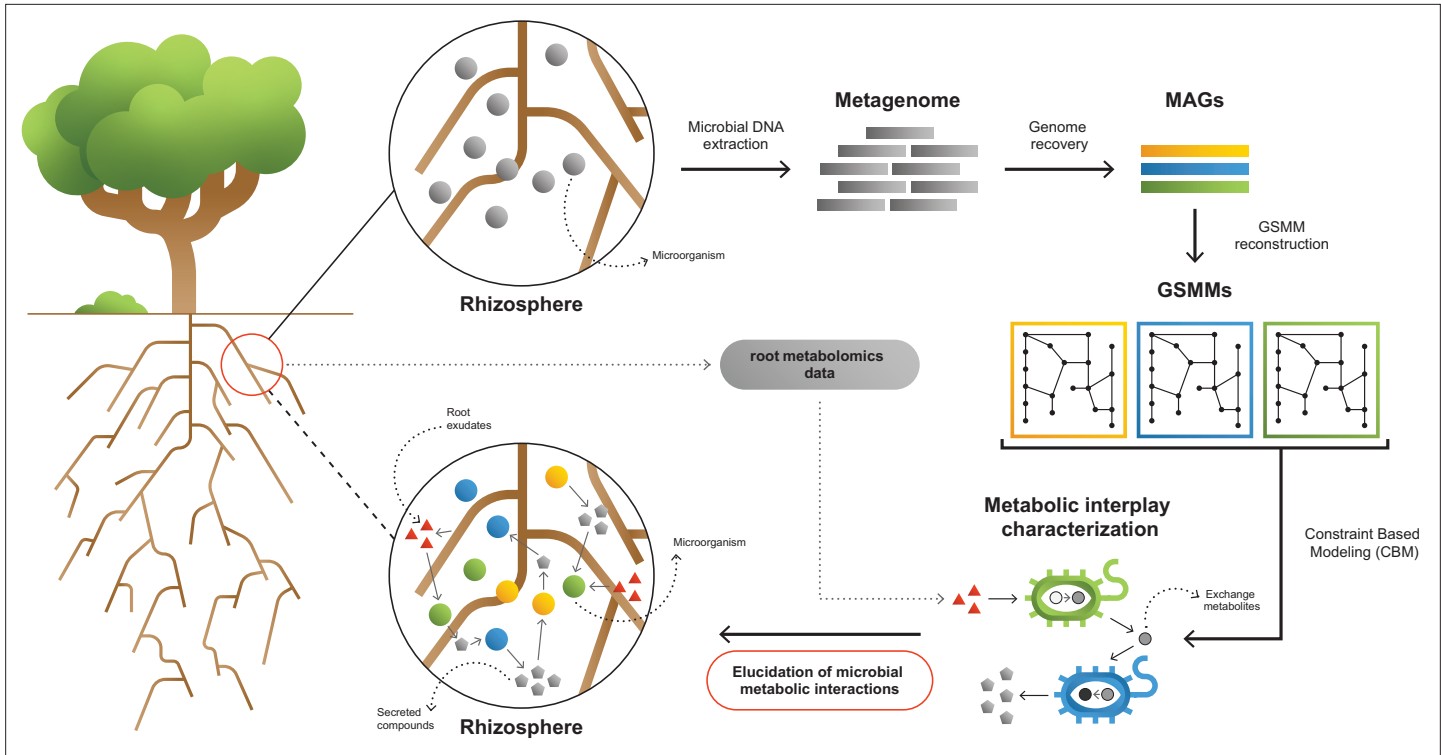

**Figure 1.** A framework for interpreting and characterizing the network of metabolic interactions of root-associated bacteria. Going from obscure rhizosphere (upper circle) to the elucidated rhizosphere (lower circle): Microbial genomic DNA extracted from rhizosphere soil serves the construction of a rhizosphere-derived metagenome and the recovery of rhizosphere community metagenome assembled genomes (MAGs). For each MAG, a genome scale metabolic model (GSMM) is built, representing a specific species in the microbial community. Apple-root exudate profiles based on metabolomics data are used for the construction of a simulation environment. Via constraint-based modeling (CBM), applied on GSMMs, interactions are characterized in the simulated root environment, yielding predictive information regarding potential trophic exchanges within the native rhizosphere microbial community.

*2020*; *Xu et al., 2019*; *Dhakar et al., 2022*). Genome recovery, or genome-resolved metagenomes, often referred to as metagenome assembled genomes (MAGs) allows one to obtain full genomes directly from metagenomes (*Taş et al., 2021*; *Uritskiy et al., 2018*). Constructing GSMMs based on MAGs derived from a specific biological sample or directly from a native community enables a genuine view of the metabolic activities carried out *in situ*, hence bypasses the need in 16S-based genome imputation. Such an in silico representation of a native community (with respect to its environment) can be used to decrypt the myriad interconnected uptake and secretion exchange fluxes transpiring within the root-associated microbiome, spanning from root exudates to altered organic forms.

The current study describes the recovery of 395 unique MAGs from metagenomes constructed for the native rhizosphere community of apple rootstocks cultivated in orchard soil affected by apple replant disease (ARD) (*Somera et al., 2021*; *Berihu et al., 2023*). Soils were amended with *Brassicaceae* seed meal, or were not amended, supporting the development of either disease-suppressive or disease-conducive root microbiomes, respectively (*Somera et al., 2021*; *Berihu et al., 2023*). MAGs were recovered from metagenomics data collected from these apple rhizosphere microbiomes. GSMMs constructed for the MAGs provide an in silico representation of highly abundant species in the native rhizosphere community. CBM simulations were then conducted in a rhizosphere-like environment, where microbial uptake-secretion fluxes were connected to form a directional trophic network. The aims of this study were twofold: first, to provide a general framework for delineating inter-species interactions occurring in the rhizosphere environment (*Figure 1*) and second, to characterize the metabolic roles specific groups of bacteria fulfill in seed meal-amended (disease-suppressive) vs non-amended (disease-conducive) apple rhizobiome communities.

## Results and discussion

### Assembly of a collection of MAGs representing a native microbial community from a soil agroecosystem

Metagenomic sequencing obtained from the rhizosphere of apple rootstocks grown in orchard soil with a documented history of replant disease resulted in a total of approximately 2 billion quality reads (after filtration) at a length of 150 bp, as described in *Berihu et al., 2023*. Independent metagenomics assemblies of six different treatments yielded 1.4–2 million contigs longer than 2 kbp (*Berihu et al., 2023*). Here, assemblies were binned using MetaWRAP (*Uritskiy et al., 2018*) into 296–433 high-quality MAGs for each of the six treatments (*Figure 2—source data 1*); completion and contamination thresholds were set to 90/5, respectively (*Bowers et al., 2017*). De-replication (the process of combining highly similar genomes into a single representative genome) requiring 99% average nucleotide identity of the overall 2233 MAGs yielded a collection of 395 unique MAGs (*Figure 2*). Across samples, 30–36% of the raw reads were mapped to the MAG collection (*Figure 2—source data 2*), in comparison to 61–71% mapped to the non-binned contigs (*Berihu et al., 2023*). Using GTDB-Tk (*Chaumeil et al., 2019*), taxonomic annotations were assigned at the phylum, order, genus, and species level for 395, 394, 237, and three of the de-replicated MAGs, respectively, reflecting the genuine diversity of the rhizosphere community, which include many uncharacterized species (*Buée et al., 2009*). Estimates of completion and contamination, total bin length and taxonomic affiliation for the MAG collections derived from each of the six assemblies, as well as the de-replicated MAGs, are provided in *Figure 2—source data 1*.

As in previous reports (*Zhalnina et al., 2018*; *Xu et al., 2018*), Proteobacteria, Acidobacteria, Actinobacteria, and Bacteroidetes were identified as the dominant phyla in the apple rhizosphere (*Figure 2A*). Overall, the taxonomic distribution of the MAG collection corresponded with the profile reported for the same samples using alternative taxonomic classification approaches such as 16S rRNA amplicon sequencing and gene-based taxonomic annotations of the non-binned shotgun contigs (*Figure 2B*). At the genus level, MAGs were classified into 143 genera in comparison to approximately 3000 genera that were identified for the same data based on gene-centric approaches (*Berihu et al., 2023*) and approximately 1000 genera based on amplicon sequencing (*Somera et al., 2021*) of the same data.

The functional capabilities of the bacterial genomes in the apple rhizosphere were initially assessed based on KEGG functional annotations of their gene catalogue (*Figure 2—source data 3*). The 10 most frequent functional categories across the MAG collection were involved in primary metabolism, for example, carbohydrate metabolism, and the biosynthesis of essential cellular building blocks such as amino acids and vitamins. Specialized functional categories included those associated with autotrophic nutrition such as carbon fixation and metabolism of nitrogen, sulfur, and methane. Functional diversity was found to exist also when considering ubiquitous functions. For instance, though all bacteria are in need of the full set of amino acids, most genomes lack the full set of relevant biosynthesis pathways. The prevalence of biosynthetic pathways across MAGs (requiring at least a single relevant enzyme) ranges from 99.5% (e.g., glycine; missing in only two genomes) to 21% (e.g., tyrosine; detected in only 83 MAGs). Notably, the diversity of metabolic pathway completeness regarding amino acids and other essential cellular components suggests that the majority of bacterial soil species rely on an external supply of at least some of their obligatory nutritional demands.

### Construction of a simulation system for exploring environment-dependent metabolic performances and growth of rhizosphere bacteria

Categorical classifications of discrete gene entities, such as pathway completeness analyses, have several inherent limitations as an approach for the contextualization and functional interpretation of genomic information. First, categorical classifications may underestimate the completeness of robust pathways with multiple redundant routes resulting in low pathway completeness. Second, pathway completeness analyses do not take into consideration the directionality and continuity of a biosynthetic process whose full conductance requires the availability of a specific environmental resource together with the required successive series of genes/reactions. Although functional potential can be inferred based on the static set of genes and enzymes present, actual metabolic performances of

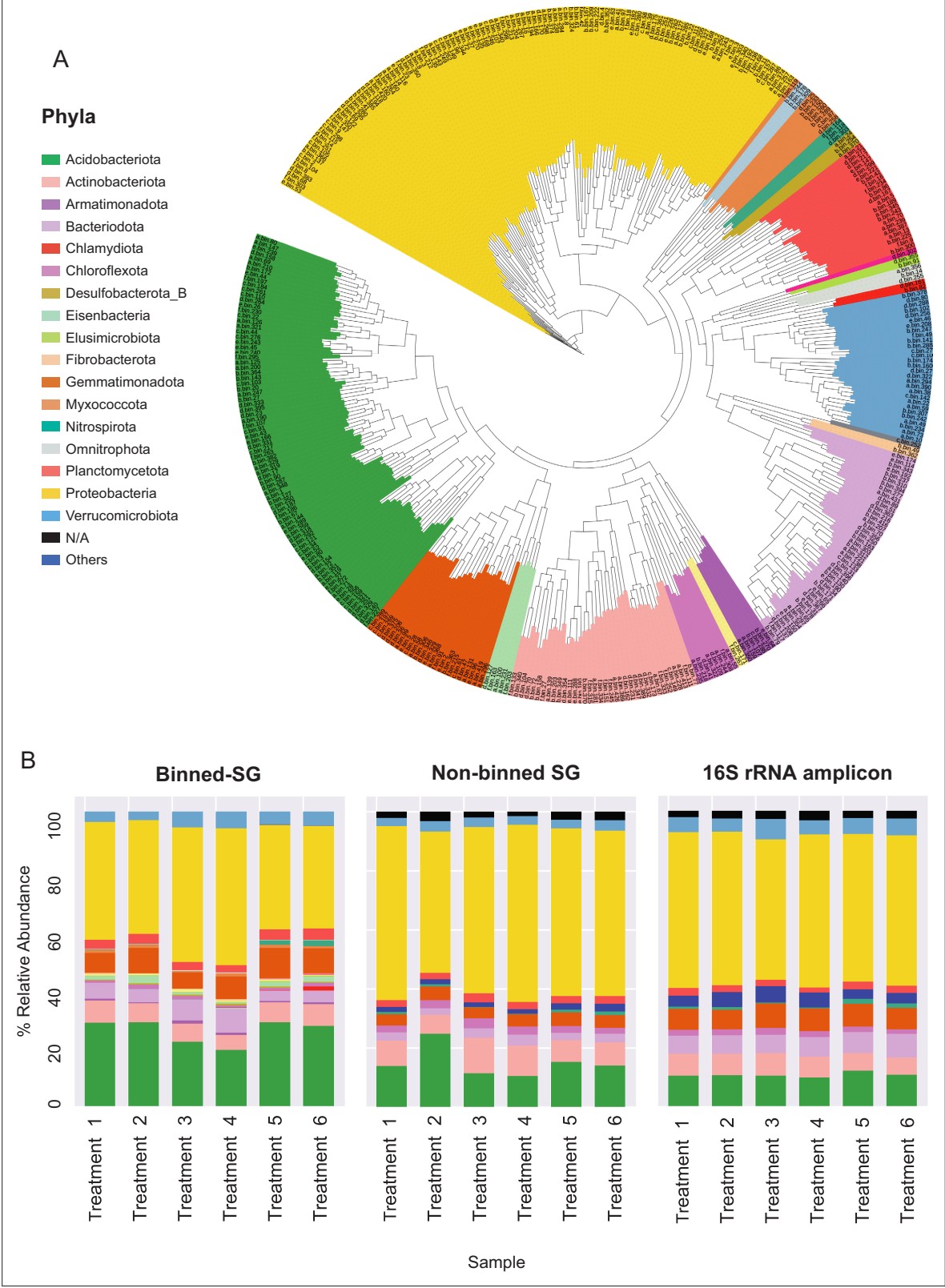

**Figure 2.** Phylogenomic surveys of the genomic collection assembled from sequence data derived from apple rhizosphere samples. (**A**) A phylogenomic tree constructed from 395 de-replicated metagenome assembled genomes (MAGs) extracted from the metagenomes. The tree is based on concatenated marker proteins according to GTDB-Tk (*Chaumeil et al., 2019*). (**B**) Relative abundance of bacterial taxa at the phylum level as inferred from different phylogenomic classification approaches applied for the same samples: Binned-SG (MAGs derived from shotgun metagenomics);

*Figure 2 continued on next page*

*Figure 2 continued*

non-binned SG (contigs derived from shotgun metagenomics); 16S rRNA amplicon derived from the same samples. Taxonomic classifications of rRNA amplicon data and non-binned-SG sequences are taken from *Somera et al., 2021* and *Berihu et al., 2023*, respectively. Shotgun metagenomics data from Berihu et. al was used here for construction of MAGs catalogue. 'Treatments 1–6' relate to six specific combinations of rootstock and soil amendment as detailed in *Figure 2—source data 2*. Relative abundance was calculated as the average value in five replicates conducted for each treatment.

The online version of this article includes the following source data and figure supplement(s) for figure 2:

**Source data 1.** Genomic characteristics of metagenome assembled genomes (MAGs).

**Source data 2.** Summary table of the mapping of reads to metagenome assembled genomes (MAGs) across treatments.

**Source data 3.** Pathway completion analysis (Kyoto Encyclopedia of Genes and Genomes [KEGG] pathway decoder) of metagenome assembled genomes (MAGs).

**Source data 4.** Genome scale metabolic model (GSMM) MEMOTE tests results.

**Source data 5.** Genome scale metabolic model (GSMM) attributes.

**Figure supplement 1.** Metabolic pathway completion analysis of community de-replicated metagenome assembled genomes (MAGs).

**Figure supplement 2.** Distribution of genome scale metabolic model (GSMM) attributes at the phylum level.

bacterial species in soil are dynamic and reflect multiple factors, including the availability of different nutritional sources. Such sources can be environmental inputs like root exudates or downstream exchange metabolites secreted by cohabiting bacteria.

To better understand environment-dependent metabolic activity occurring in a native rhizosphere, a set of 395 GSMMs was constructed for the entire collection of the rhizosphere-bacterial community MAGs. All models were systematically subjected to validity and quality tests using MEMOTE (*Lieven et al., 2020*), leaving a total of 243 GSMMs whose stoichiometric consistency was confirmed (*Figure 2—source data 4*). On average, GSMMs included 1924 reactions, from which 203 were exchange reactions (specific reactions carrying out extracellular import and secretion of metabolites) and 1312 metabolites (*Figure 2—source data 5*). Altogether, the GSMM set held 5152 unique metabolic reactions, 597 exchange reactions, and 2671 different compounds. The distribution of key model attributes (reactions, metabolites, and exchanges) across phyla is shown in *Figure 2—figure supplement 2*. Model features were scalable with those reported by *Basile et al., 2020* and *Heinken et al., 2023*. Additionally, comparison of GSMM scales indicated that the metabolic coverage (i.e., the number of reactions, which denote the potential of executing a metabolic function) of our data is within the same order of magnitude as described in recent large-scale automatic reconstructions (*Basile et al., 2020*; *Heinken et al., 2023*).

Next, species-specific rich (optimal) and poor (suboptimal) media were defined for each model in order to broadly assess GSMM growth capacity. A rich environment was defined as a medium containing all metabolites for which the model encodes an exchange reaction. A poor environment was defined as the minimal set of compounds enabling growth (see Methods). Essentially, poor environments contained species-specific carbon, nitrogen and phosphorous sources together with other trace elements. In addition to the two automatic media, we aimed to design a realistic simulating

**Table 1.** Apple-root exudates adopted from the works of *Leisso et al., 2017*; *Leisso et al., 2018*, included in the rhizosphere environment.

| Category | Compounds |
| --- | --- |
| Amino acids | L-Asparagine, L-aspartate, L-cysteine, L-valine, beta-alanine |
| Monosaccharides | Rhamnose, glycerate, ribose, galactose, xylose, erythrose |
| Sugar-alcohols | Sorbitol, galactitol, glycerol |
| Other carbohydrates | 3,4-Dihydroxybenzoate, 3,4-dihydroxy-trans-cinnamate, 4-hydroxybenzoate, benzoate, esculin, ferulate, gallic acid, quinate, salicin, trans-cinnamate |
| Organic acids | D-Galacturonate, D-lactate, D-malate, malonate, oxalate, pyruvate, succinate |
| Carbamides | Urea |
| Fatty acyls | Octadecanoic acid |

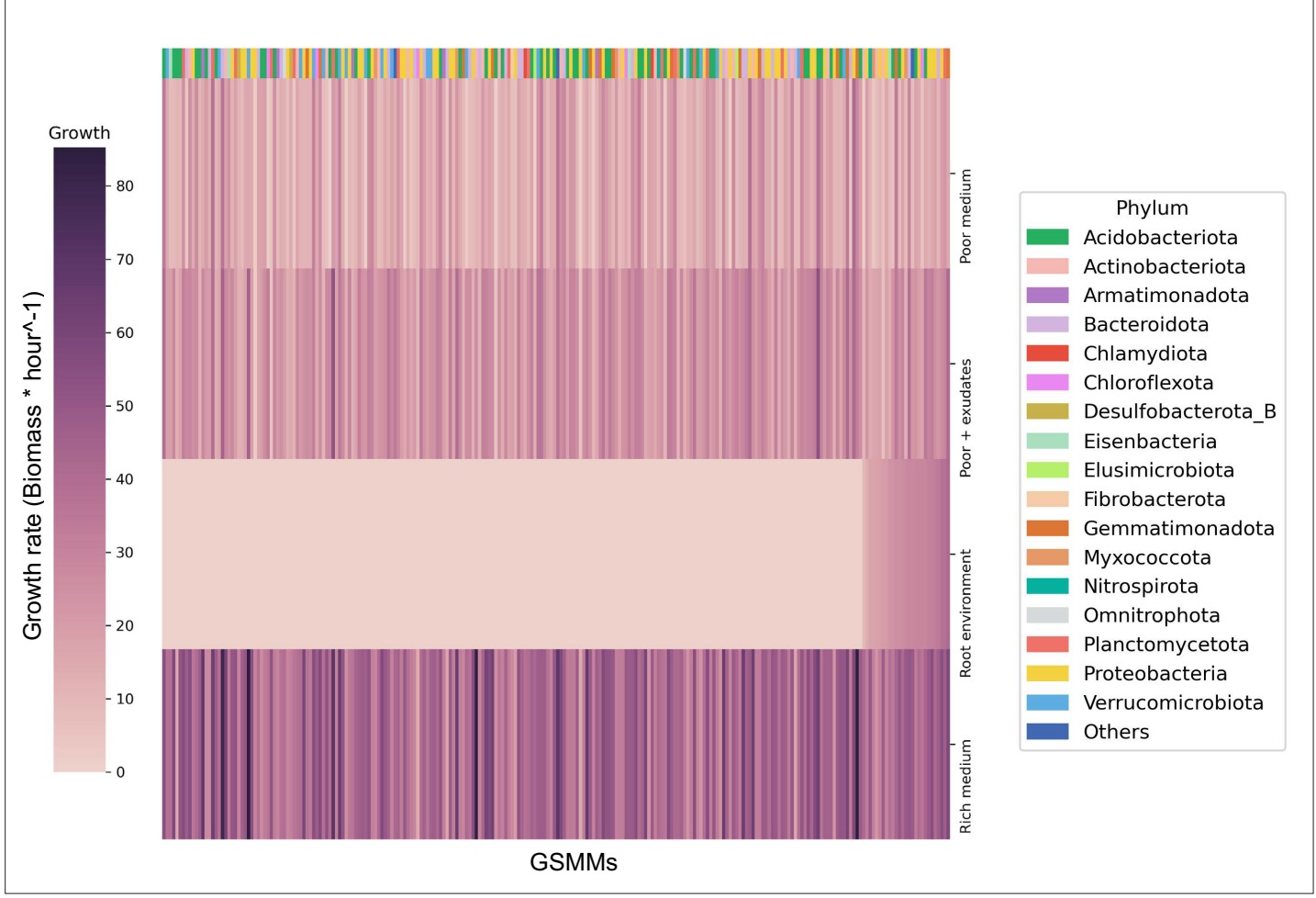

**Figure 3.** Bacterial growth rates in different environments. Each row represents the growth rate (biomass increase * hour⁻¹) of species in a specialized medium; from top to bottom: poor medium, poor medium supplemented with root exudates, root environment medium, and rich medium. Each column represents a genome scale metabolic model (GSMM). Models are sorted by their growth score in the root environment medium. Poor, rich, and roor + exudates media are model specific, thus sustain growth of all models. The horizontal color bar on top of the plot represents the phyla of the corresponding GSMM. Actual growth values are provided in *Figure 3—source data 1*.

The online version of this article includes the following source data and figure supplement(s) for figure 3:

**Source data 1.** Genome scale metabolic model (GSMM) growths in different environments.

**Figure supplement 1.** Flux Variability Analysis (FVA) performances of genome scale metabolic models (GSMMs) in different environments (*Figure 3— source data 1*).

environment to explore the impact of the root exudates on the native community. To this end, metabolomics data from a set of studies which characterized the root exudates of Geneva 935 (G935) or Malling 26 (M26) rootstock cultivars (*Leisso et al., 2017*; *Leisso et al., 2018*) – related to the G210 and M26 rootstocks whose rhizobiome was characterized here, were used to specify an array of apple root-derived compounds (*Table 1*). The list of secreted compounds was consistent with other reported profiles of plant root exudates (*Zhalnina et al., 2018*; *Naveed et al., 2017*).

The growth of each of the 243 GSMMs was simulated in each of the three species-specific environments (rich, poor, poor + exudates; exudates were added to corresponding poor media to ensure the exudates have a feasible effect). As expected, growth performances were higher on the poor medium supplemented with exudates in comparison to growth on the poor medium alone but lower than the growth in rich medium (*Figure 3*). Notably, GSMM growth patterns were not phylogenetically conserved and were inconsistent between related taxa. Moreover, ranking of models' growth rate was inconsistent in the three different media (i.e., some models' growth rates were markedly affected by the simulated media whereas others did not; *Figure 3*). This inconsistency indicates that the effect of

exudates on community members is selective and differs between species (i.e., exudates increase the growth rates of some species more than others), as was previously reported (*Zhalnina et al., 2018*; *Sasse et al., 2018*; *Stringlis et al., 2018*). The number of active exchange fluxes in each medium corresponds with the respective growth performances displaying noticeably higher number of potentially active fluxes in the rich enviroenment (also when applying loopless Flux Variability Analysis [FVA]) (*Figure 3—figure supplement 1*). Overall, simulations confirmed the existence of a feasible solution space for all the 243 models as well as their capacity to predict growth in the respective environemnt (*Figure 3—source data 1*).

As a next step toward conducting simulations in a genuine natural-like environment, we aimed to define a single 'rhizosphere environment' in which growth simulations for all models would take place. Unlike the species-specific root media (poor medium + exudates) which support growth of all models by artificially including multiple carbon sources that are derived from the automatic specifications of the poor medium, including such that are not provided by the root, this simulation environment was based on the root exudates (*Table 1*) as the sole carbon sources. By avoiding the inclusion of non-exudate organic metabolites, the true-to-source rhizosphere environment was designed to reveal the hierarchical directionality of the trophic exchanges in soil, as rich media often mask various trophic interactions taking place in native communities (*Opatovsky et al., 2018*). Additionally, the rhizosphere environment also included an array of inorganic compounds used by the 243 GSMMs, which includes trace metals, ferric, phosphoric, and sulfuric compounds. Overall, the rhizosphere environment was composed of 60 inorganic compounds together with the 33 root exudates (*Figure 4—source data 1*). The rhizosphere environment supported the growth of only a subset of the GSMMs that were capable of using plant exudates (*Figure 3*).

## Simulating growth succession and hierarchical trophic exchanges in the rhizosphere community

To reflect the indirect effect of the root on the native community (i.e., to capture the effect of root-supported bacteria on the growth of further community members), we constructed the microbial community succession module (MCSM), a CBM-based algorithm aimed at predicting community-level trophic successions. MCSM utilizes FVA to simulate and enumerate the exchange fluxes of individual models, extending their secretion profiles beyond the standard FBA-based solutions commonly used in other CBM tools designed for modeling microbial interactions (*Basile et al., 2020*; *Pacheco et al., 2019*; *San Roman and Wagner, 2018*; *Diener et al., 2020*; *Dukovski, 2021*). Unlike certain CBM tools designed for modeling microbial community interactions (*Diener et al., 2020*), MCSM bypasses the need to define a community objective function, as the growth of each species is simulated individually. Trophic interactions are inferred by the extent to which exchange compounds secreted by bacteria could support the growth of other community members. MCSM iteratively grows the GSMMs in a defined environment, sums up their individual secretion profiles, and updates the initial simulation environment with those secreted compounds (*Figure 4A*). Applying this algorithm to a microbiome in its native environment allows delineating the potential metabolic dependencies and interactions between bacterial species in a native community.

Application of MCSM over the 'rhizosphere environment' (i.e., first iteration, root exudates, and 'inorganic compounds', *Figure 4—source data 1*) supported the growth of 27 GSMMs (*Figures 3 and 4B*). Then, the initial environment was updated with 145 additional compounds predicted to be secreted by the growing community members. The second iteration supported the growth of 33 additional species whose growth was supported by compounds predicted to be secreted by bacteria that grew in the first iteration. Following the second iteration, 25 new secreted compounds were added to the rhizosphere environment. The third iteration supported the growth of 11 additional GSMMs, with one additional compound secreted. After the third iteration, the updated environment did not support the growth of any new species. Overall, iterative growth simulations resulted in the successive growth of 71 species (*Figure 4B*; iterations 1–3).

To enlarge the array of growing species, we tested the effect of the addition of organic phosphorous sources. Organic phosphorous is typically a limiting factor in soil (*Huang et al., 2017*) and its utilization varies greatly between microbial species [i.e., different P sources were shown to have a selective effect on different microbial groups (*Zheng et al., 2019*)]. The initial rhizosphere environment contained only inorganic phosphorous. During first to third MCSM simulations, nine organic P

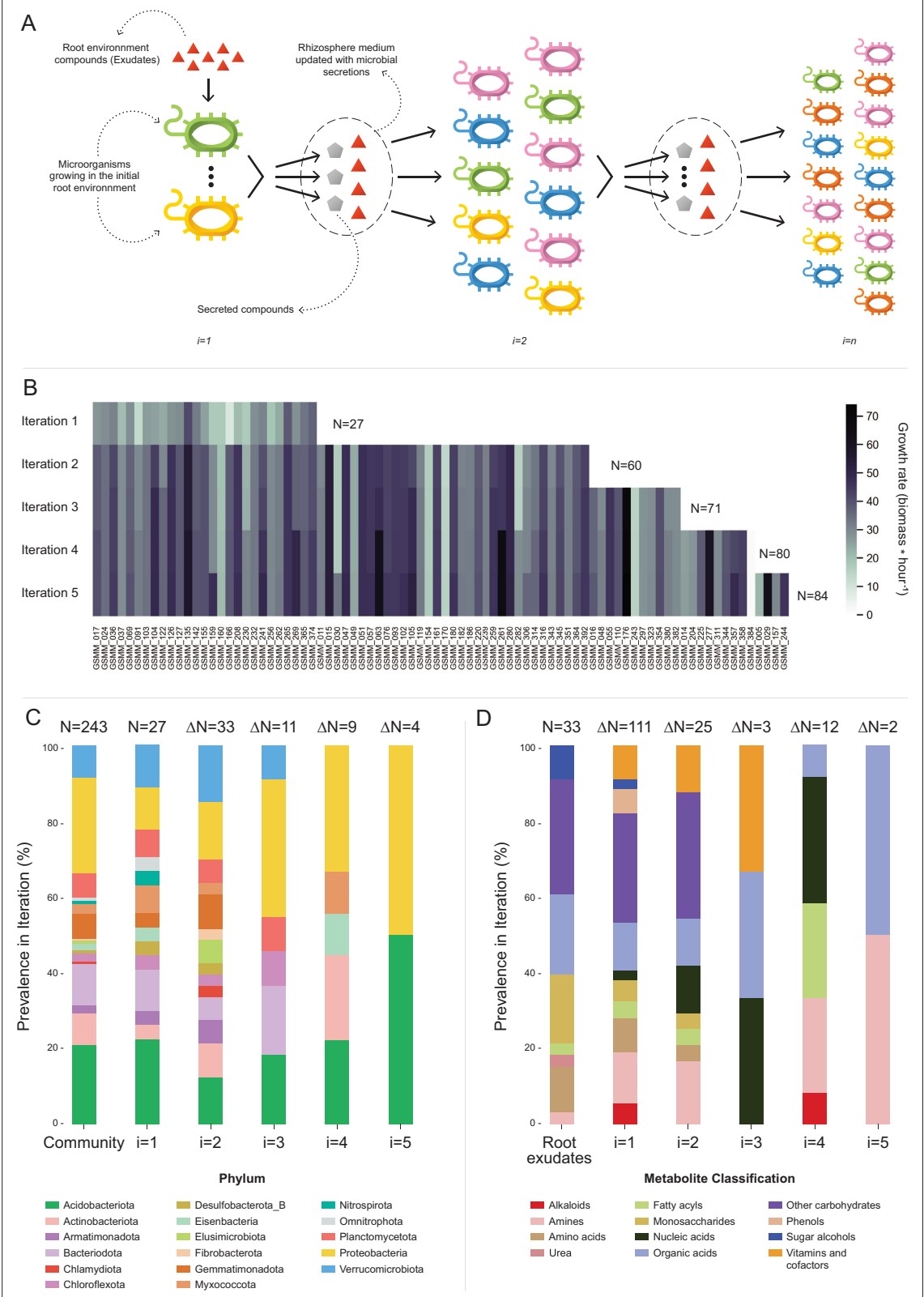

**Figure 4.** Microbial community succession module (MCSM), and characterization of trophic dynamics in the community along iterations. (**A**) An illustration of the iterative microbial community growth module, representing the growth of community members along iterations starting in the 'Rhizosphere environment' and updated with microbial secretion outputs. (**B**) Growth rate characterization of genome scale metabolic models (GSMMs) in the community along iterations. Each column in the heat-map represents a different GSMM (only models which have grown in the rhizosphere

*Figure 4 continued on next page*

*Figure 4 continued*

environment are presented); growth rate is indicated by the color bar. Iterations are represented by rows. Blank spaces indicate models not growing at that specific iteration. N is the total number of species that grew after each iteration. (**C**) Distribution of GSMMs growing along iterations at the phylum level. (**D**) Distribution of organic compounds secreted along iterations, classified into biochemical groups. Root exudates bar (far left) represents the classification of organic compounds in the initial 'Rhizosphere environment'. Numbers on top of bars in both C and D (designated by N) denote the number of new entries in a specific iteration, with respect to the previous iteration.

The online version of this article includes the following source data and figure supplement(s) for figure 4:

**Source data 1.** Table of compounds used throughout iterations in the microbial community succession module (MCSM).

**Source data 2.** Genome scale metabolic model (GSMM) secretion data derived from microbial community succession module (MCSM).

**Source data 3.** Initial medium compounds used in microbial community succession module (MCSM) cellulose degradation process.

**Figure supplement 1.** Phylogenetic distribution of models growing along iterations at the order level.

**Figure supplement 2.** Uptake and secretion degrees of biochemically classified metabolite nodes, as inferred from the communal interaction network.

**Figure supplement 3.** Application of microbial community succession module (MCSM) over the process of cellulose decomposition as described by *Kato et al., 2005*.

compounds were secreted to the simulation medium, which was updated accordingly. At the beginning of the fourth iteration, 31 additional organic P compounds were identified by screening the species-specific poor medium and were added to the medium (*Figure 4—source data 1*; organic phosphorous compounds). The additional organic P compounds supported the growth of nine additional GSMMs (*Figure 4B*) and led to the secretion of 13 new compounds, which were added to the environment. The fifth iteration supported the growth of four additional GSMMs and two new compounds were secreted. Final simulations in the cumulative rhizosphere environment were composed of all secreted compounds and led to the same secretion and growth profile as the previous iteration. Therefore, no further growth iterations were conducted.

Overall, the successive iterations connected 84 out of 243 native members of the apple rhizosphere GSMM community via trophic exchanges. The inability of the remaining bacteria to grow, despite being part of the native root microbiome, possibly reflects the selectiveness of the root environment, which fully supports the nutritional demands of only part of the soil species, whereas specific compounds that might be essential to other species are less abundant (*Buée et al., 2009*). It is important to note that the specific exudate profile used here represents a snapshot of the root metabolome as root secretion profiles are highly dynamic, reflecting both environmental and plant developmental conditions. A possible complementary explanation to the observed selective growth might be the partiality of our simulation platform, which examined only plant–bacteria and bacteria–bacteria interactions while ignoring other critical components of the rhizosphere system such as fungi, archaea, protists, and mesofauna, as well as less abundant bacterial species, components all known to metabolically interact (*Bardgett and Putten, 2014*). Finally, the MAG collection, while relatively substantial, represents only part of the microbial community. Accordingly, the iterative growth simulations represent a subset of the overall hierarchical trophic exchanges in the root environment, necessarily reflecting the partiality of the dataset.

In terms of the phylogenetic distribution of the models, 27 bacterial species grew on the first iteration (in which root exudates served as the sole organic sources). These bacteria represented 14 of the 17 phyla included in the initial model collection (consisting of 243 GSMMs) and maintained a distribution frequency similar to the original community. As in the full GSMM dataset (Community bar, *Figure 4C*), most of the species which grew in the first iteration belonged to the phyla *Acidobacteriota*, *Proteobacteria*, and *Bacteroidota*. This result concurred with findings from the work of Zhalnina et al., which reported that bacteria assigned to these phyla are the primary beneficiaries of root exudates (*Zhalnina et al., 2018*). Species from 3 out of the 17 phyla that did not grow in the first iteration – *Elusimicrobiota*, *Chlamydiota*, and *Fibrobacterota*, did grow on the second iteration (*Figure 4C*). Members of these phyla are known for their specialized metabolic dependencies. Such is the case for example with members of the *Elusimicrobiota* phylum, which include mostly uncultured species whose nutritional preferences are likely to be selective (*Uzun et al., 2023*).

At the order level, bacteria classified as *Sphingomonadales* (class *Alphaproteobacteria*), a group known to include typical inhabitants of the root environment (*Lei et al., 2019*), grew in the initial root environment. In comparison, other root-inhabiting groups including the orders *Rhizobiales* and

*Burkholderiales* (*Lei et al., 2019*), did not grow in the first iteration. *Rhizobiales* and *Burkholderiales* did, however, grow in the second and third iterations, respectively, indicating that in the simulations, the growth of these groups was dependent on exchange metabolites secreted by other community members (*Figure 4—figure supplement 1*).

Overall, 158 organic compounds were secreted throughout the MCSM simulation (from which 12 compounds overlapped with the original exudate medium). These compounds varied in their distribution and were mapped into 12 biochemical categories (*Figure 4D*). Whereas plant secretions are a source of various organic compounds, microbial secretions provide a source of multiple vitamins and co-factors not secreted by the plant. Microbial-secreted compounds included siderophores (staphyloferrin, salmochelin, pyoverdine, and enterochelin), vitamins (pyridoxine, pantothenate, and thiamin), and coenzymes (coenzyme A, flavin adenine dinucleotide, and flavin mononucleotide) – all known to be exchange compounds in microbial communities (*Ghosh et al., 2017*; *Lu et al., 2020*). In addition, microbial secretions included 11 amino acids (arginine, lysine, threonine, alanine, serine, phenylalanine, tyrosine, leucine, glutamate, isoleucine, and methionine), also known as a common exchange currency in microbial communities (*Mee et al., 2014*). Some microbial-secreted compounds, such as phenols and alkaloids, were reported to be produced by plants as secondary metabolites (*Justin et al., 2014*; *Yang et al., 2022*). Additional information regarding mean uptake and secretion degrees of compounds classified to biochemical groups is found in *Figure 4—figure supplement 2*.

Conceptually, the rhizosphere microbiota can be classified into two trophic groups: primary exudate consumers, comprising microbial species that are direct beneficiaries from the root exudates, and secondary consumers, comprising microbial species whose growth may be provided directly via the uptake of metabolites secreted by other members of the soil microbial community. In the iterative MCSM simulations, compounds secreted by some of the primary consumers largely sustained the growth of secondary consumers, which were not able to grow otherwise. The full information on the secretion profiles and models' growths is provided in *Figure 4—source data 2*.

To validate the ability of MCSM to capture trophic dependencies and succession, we further tested whether it can track the well-documented example of cellulose degradation – a multi-step process conducted by several bacterial strains that go through the conversion of cellulose and its oligosaccharide derivatives into ethanol, acetate, and glucose, which are all eventually oxidized to $CO_2$ (*Kato et al., 2005*). Here, the simulation followed the trophic interactions in an environment provided with cellulose oligosaccharides (4 and 6 glucose units) on the first iteration (*Figure 4—source data 3*). The formed trophic successions detected along iterations captured the reported multi-step process (*Figure 4—figure supplement 3*).

## Associating trophic exchanges with soil health

The MAG collection analyzed in this study was constructed from shotgun libraries associated with apple rootstocks cultivated in orchard soil with a documented history of ARD and healthy/recovered (seed meal-amended) soils, providing a model system for disease-conducive vs disease-suppressive rhizosphere communities (*Somera et al., 2021*). Briefly, rhizobiome communities obtained from apple rootstocks grown in replant orchard soil leading to symptomatic growth (non-amended samples) were termed 'sick', whereas samples in which disease symptoms were ameliorated following an established soil amendment treatment (*Mazzola et al., 2015*), were termed 'healthy'. Both sick and healthy plants were characterized by distinct differences in the structure and function of their rhizosphere microbial communities in the respective soil samples (*Mazzola and Freilich, 2017*; *Somera et al., 2021*; *Berihu et al., 2023*; *Mazzola et al., 2015*). In order to correlate microbial metabolic interactions with soil performance, GSMMs were classified into one of three functional categories based on differential abundance (DA) patterns of their respective MAGs: predominantly associated with 'healthy' soil (H), predominantly associated with 'sick' soil (S), and none-associated (NA) (*Figure 5—source data 1*).

The functionally classified GSMMs (H, S, and NA) were consolidated into a community network of metabolic interactions by linking their potential uptake and secretion exchange profiles (as predicted along growth iterations in *Figure 4*). The network was built as a directed bipartite graph, in which the 84 feasible GSMM nodes and the 203 metabolite nodes (27 root exudates, 146 microbial-secreted compounds, and 30 additional organic-P compounds) were connected by 9773 directed edges, representing the metabolic exchanges of organic compounds in the native apple rhizosphere community (*Figure 5A*). Further information regarding node degrees is found in *Figure 5—source data 2*.

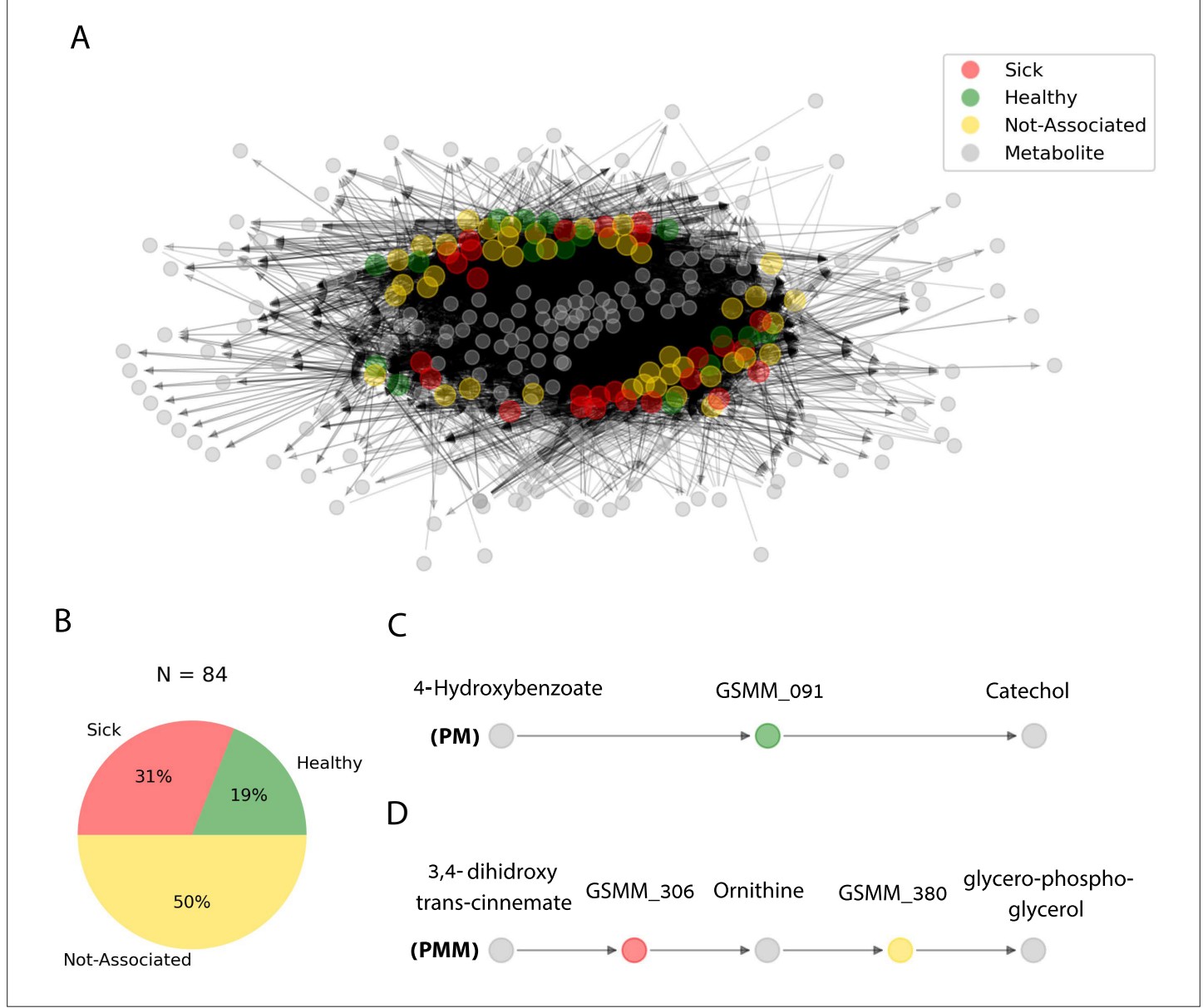

**Figure 5.** Trophic interactions based on community exchange fluxes predicted along iterations in the simulated rhizosphere environment. (**A**) Network representation of potential metabolite exchanges between rhizosphere community members. Edges in the network are directional; arrowhead from a gray node (metabolite) pointing toward a colorful node (genome scale metabolic model, GSMM) indicates uptake; arrowhead from a colorful node (GSMM) pointing toward a gray node (metabolite) indicates secretion. Node colors correspond to differential abundance classification of GSMMs in the different plots: healthy, sick, and not-associated. Metabolites found in the center of the network are of a higher connectivity degree (i.e., are involved in more exchanges). Only organic compounds are included in the network. Gray rectangles illustrate a zoom-in to specific five- and three-partite sub-networks (**C, D**). (**B**) Pie chart distribution of GSMMs classified according to differential abundance of reads mapped to the respective metagenome assembled genomes (MAGs); *N* is the total number of GSMMs in the network. (**C, D**) Examples of specific sub-network motifs derived from the community network; plant–microbe (PM; a three component sub-network motif, upper); plant–microbe–microbe (PMM; a five component sub-network motif, lower), respectively. The full list of PM and PMM sub-network motifs is found in *Figure 6—source data 1*.

The online version of this article includes the following source data for figure 5:

**Source data 1.** Differential abundance scores of genome scale metabolic models (GSMMs) (metagenome assembled genomes, MAGs).

**Source data 2.** Trophic network information, nodes degree.

The directionality of the network enabled its untangling into sub-network motifs stemming from a root exudate to exchange interactions, and ending with an unconsumed end-metabolite. Two types of sub-networks were detected (*Figure 5C, D*): 3-component (PM) plant exudate–microbe–microbial-secreted metabolite; and 5-component (PMM; plant–microbe–microbe) plant exudate–microbe–intermediate microbial-secreted metabolite–microbe–microbial-secreted metabolite. Overall, the network included 45,972 unique PM paths and 571,605 unique PMM paths. Participation of GSMMs in PM paths ranged from 272 to 896 occurrences (*Figure 6—figure supplement 1A*). GSSM participation in PMM paths ranged from 398 to 50,628 in the first microbe position (primary exudate consumer) and 1388–19,738 occurrences in the second microbe position (secondary consumer) (*Figure 6—figure supplement 1B, C*). Frequency of GSMMs in the first position in PMM sub-network motifs was negatively correlated with the frequency of presence in second positions, possibly indicating species-specific preferences for a specific position/trophic level in the defined environment (Pearson = −0.279; p-value = 0.009, *Figure 6—figure supplement 1B–D*).

In order to explore the trophic preferences of bacteria associated with the different rhizosphere soil systems, the frequency of healthy (H), sick (S), or non-associated (NA) GSMMs in the PM and PMM sub-networks was compared (*Figure 6—source data 1*). GSMMs classified as S initiated a significantly higher number of PMM sub-networks (located in the first position) than GSMMs classified as NA and H (*Figure 6A*). H-classified PMM paths (first position) initiated a significantly higher number of sub-networks with GSMMs classified as NA compared to S-classified GSMMs, but no more than H-classified GSMMs (second position). Other PMM types did not show a significant effect at the second position. The higher number of trophic interactions formed by the S-classified primary exudate consumers in the PMM sub-network motifs suggests that non-beneficial bacteria may have a broader spectrum in terms of their utilization potential of root-secreted carbon sources compared to plant-beneficial bacteria. This might shed light on the dynamics of ARD, in which S-classified bacteria become increasingly dominant following long-term utilization of apple-root exudates, resulting in diminished capacity of the rhizosphere microbiome to suppress soil-borne pathogens (*Mazzola et al., 2015*; *Mazzola, 1999*).

In order to predict exchanges with potential to support/suppress dysbiosis, the frequency of DA GSMM types (i.e., H or S) associated with metabolites (either consumed or secreted) in the PM paths was assessed (*Figure 6—source data 2*). Considering consumed metabolites (root exudates), three and six compounds were found to be significantly more prevalent in H- and S-classified PM paths, respectively (*Figure 6B*). Notably, the S-classified root exudates included compounds reported to support dysbiosis and ARD progression. For example, the S-classified compounds gallic acid and caffeic acid (3,4-dihidroxy-trans-cinnamate) are phenylpropanoids – phenylalanine intermediate phenolic compounds secreted from plant roots following exposure to replant pathogens (*Balbín-Suárez et al., 2021*). Though secretion of these compounds is considered a defense response, it is hypothesized that high levels of phenolic compounds can have autotoxic effects, potentially exacerbating ARD. Additionally, it was shown that genes associated with the production of caffeic acid were upregulated in ARD-infected apple roots, relative to those grown in γ-irradiated ARD soil (*Weiß et al., 2017b*; *Weiß et al., 2017a*), and that root and soil extracts from replant-diseased trees inhibited apple seedling growth and resulted in increased seedling root production of caffeic acid (*Sun et al., 2022*).

As to the microbial-secreted compounds, a total of 79 unique compounds were found to be significantly overrepresented in either S (42 compounds) or H (41 compounds) classified PM paths (*Figure 6C, D*). Several secreted compounds classified as healthy exchanges (H) were reported to be potentially associated with beneficial functions. For instance, the compounds L-sorbose (EX_srb__L_e) and phenylacetaladehyde (EX_pacald_e), both over-represented in H paths (*Figure 6C*), have been shown to inhibit the growth of fungal pathogens associated with replant disease (*Howell, 1978*; *Zou et al., 2007*). Phenylacetaladehyde has also been reported to have nematicidal qualities (*Gomes et al., 2020*).

Combining both exudate uptake data and metabolite secretion data, the full H-classified PM path 4-hydroxybenzoate; GSMM_091; catechol (*Figure 5C*; the consumed exudate, the GSMM, and the secreted compound, respectively) provides an exemplary model for how the proposed framework can be used to guide the design of strategies which support specific, advantageous exchanges within the rhizobiome. The root exudate 4-hydroxybenzoate is metabolized by GSMM_091 (class *Verrucomicrobiae*, order *Pedosphaerales*) to catechol. Catechol is a precursor of a number of catecholamines,

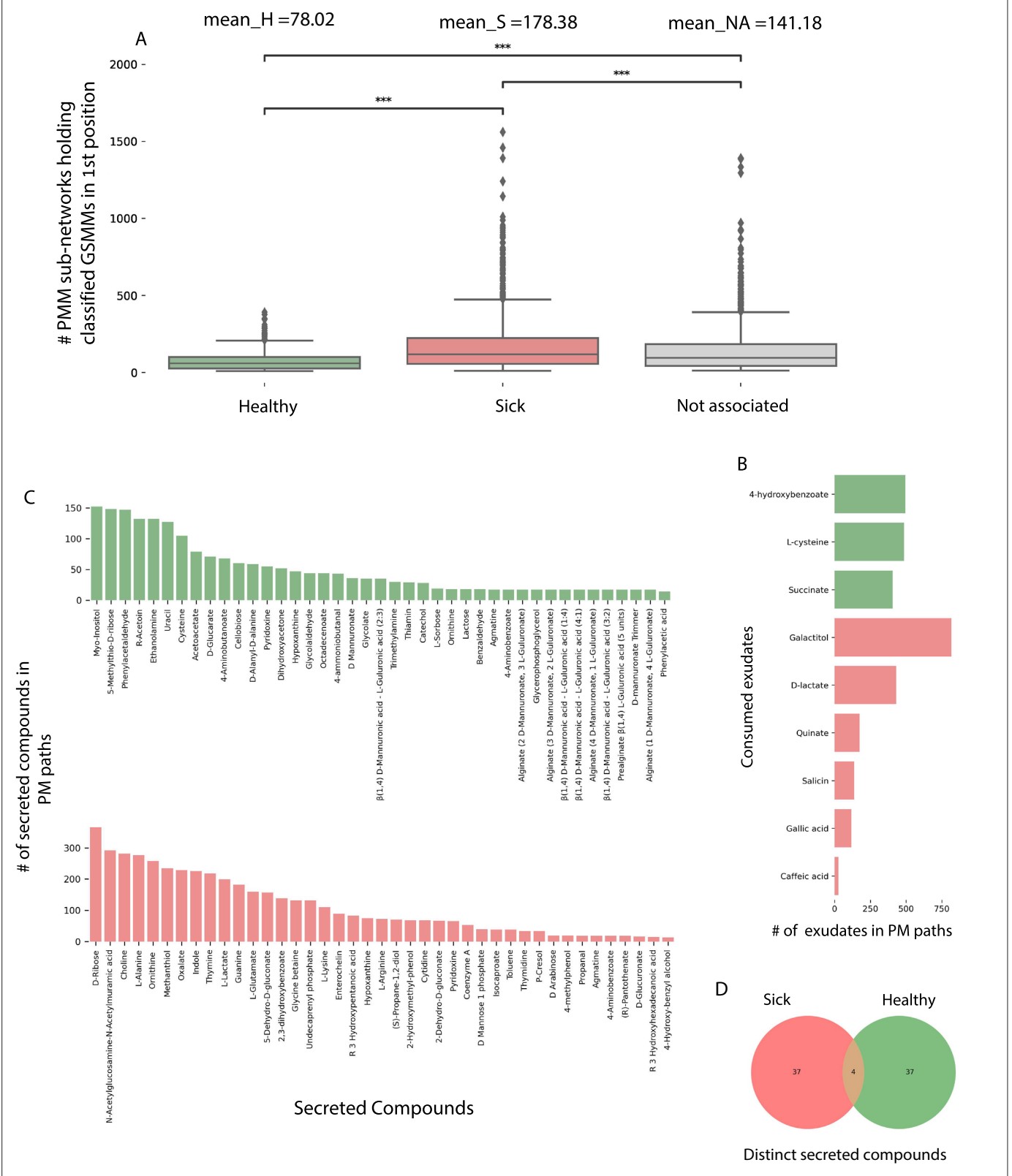

**Figure 6.** Characterization of plant–microbe (PM) and plant–microbe–microbe (PMM) sub-network motifs' features associated with differentially abundant (DA) genome scale metabolic models (GSMMs). (**A**) Count distribution of PMM sub-networks determined by GSMMs in the first microbial position classified as either healthy, sick, or not-associated, corresponding to the mean values H, S, and NA, respectively. Boxes extend from first quantile to the third quantile, middle line represents the median; dots outside whiskers indicate outliers. Distinction of groups was determined

*Figure 6 continued on next page*

*Figure 6 continued*

using ANOVA followed by the Tukey post hoc test. Asterisks indicate significance of test (***≤0.005) (**B**) Bar plot indicating the number of exudates significantly associated with H- or S-classified PM sub-networks (hypergeometric test; False Discovery Rate (FDR) ≤0.05; green: healthy – H, red: sick – S). (**C**) Bar plots indicate the number of secreted compounds in PM sub-networks, which are significantly associated with H-classified (upper, colored green) or S-classified (lower, colored red) (hypergeometric test; FDR ≤0.05). (**D**) Venn diagram represents the intersection of secreted compounds derived from both Sick and Healthy classified PM sub-networks.

The online version of this article includes the following source data and figure supplement(s) for figure 6:

**Source data 1.** Sub-networks motifs data.

**Source data 2.** Plant–microbe (PM) sub-networks functional characterization and statistical tests results.

**Figure supplement 1.** Community genome scale metabolic model (GSMM)-centric characterization of plant–microbe (PM) and plant–microbe–microbe (PMM) motifs.

a group of compounds which was recently shown to increase apple tolerance to ARD symptoms when added to orchard (*Berihu et al., 2023*; *Gao et al., 2021*). This analysis (PM; *Figure 5C*) leads to formulating the testable prediction that 4-hydroxybenzoate can serve as a selective enhancer of catecholamine synthesizing bacteria associated with reduced ARD symptoms, and therefore serve as a potential source for indigenously produced beneficial compounds.

## Conclusions

In this study, we present a framework combining metagenomics analyses with CBM, which can be used to gain a deeper understanding of the functionality, dynamics, and division of labor among rhizosphere bacteria, and link their environment-dependent metabolism to biological significance. This exploratory framework aims to illuminate the black box of interactions occurring in the rhizospheres of crop plants and is based on the work of *Berihu et al., 2023*, in which a gene-centric analysis of metagenomics data from apple rhizospheres was conducted (*Berihu et al., 2023*).

We recovered high-quality sets of environment-specific MAGs, constructed the corresponding GSMMs, and simulated community-level metabolic interactions. By including authentic apple-root exudates in the models, we were able to begin untangling the highly complex plant–bacterial and bacterial–bacterial interactions occurring in the rhizosphere environment. More specifically, we used the framework to investigate a microbial community via examining its hierarchical secretion-uptake exchanges along multiple iterations (*Figure 4*). These analyses, which linked community-derived secretion profiles with the growth of other community members, demonstrated the successive, trophic-dependent nature of microbial communities. These interactions were elucidated via construction of a community-exchange network (*Figure 5*). Possible connections between root exudates, differentially abundant (DA) bacteria, their secreted end-products, and soil health were explored using the data derived from this network. From these analyses, we were able to associate different metabolic functionalities with diseased or healthy systems, and formulate new hypotheses regarding the general function of DA bacteria in the community.

The framework we present is currently conceptual. Dealing with a highly complex system such as the rhizobiome inevitably comes with limitations. These limitations include the usage of automatic GSMM reconstruction, inherent caveats of CBM and the use of single-species GSMMs, the lack of transcriptomic and spatial-chemo-physical data, and the exclusion of competition over all its forms. Furthermore, a portion of the metabolomics data used in this framework was taken from a different source (different rootstock genotype), possibly introducing further bias to the analyses. This potential factor is due to the inherent discrepancy between the conditions from which genomics and metabolomics data were collected (*Zhalnina et al., 2018*; *Korenblum et al., 2020*). Also not considered in this framework is the role of eukaryotes in the microbial-metabolic interplay. Moreover, the use of an automatic GSMM reconstruction tool (CarveMe; *Machado et al., 2018*), though increasingly used for depicting phenotypic landscapes, is generally less accurate than manual curation of metabolic models (*Henry et al., 2010*). This approach typically neglects specialized functions involving secondary metabolism (*Freilich et al., 2011*) and introduces additional biases such as the overestimation of auxotrophies (*Price, 2023*; *Machado and Patil, 2023*). Nevertheless, manual curation is practically non-realistic for hundreds of MAGs, an expected outcome considering the volume of sequencing projects nowadays. As the primary motivation of this framework is the development of a tool capable

of transforming high-throughput, low-cost genomic information into testable predictions, the use of automatic metabolic network reconstruction tools was favored, despite their inherent limitations, in pursuit of addressing the necessity of pipelines systematically analyzing metagenomics data.

For these reasons, among others, the framework presented here is not intended to be used as a stand-alone tool for determining microbial function. The framework presented is designed to be used as a platform to generate educated hypotheses regarding bacterial function in a specific environment in conjunction with actual carbon substrates available in the particular ecosystem under study. The hypotheses generated provide a starting point for experimental testing required to gain actual, targeted, and feasible applicable insights (*Dhakar et al., 2022*; *Berihu et al., 2023*). While recognizing its limitations, this framework is in fact highly versatile and can be used for the characterization of a variety of microbial communities and environments. Given a set of MAGs derived from a specific environment and environmental metabolomics data, this computational framework provides a generic simulation platform for a wide and diverse range of future applications.

In the current study, the root environment was represented by a single pool of resources (metabolites). As genuine root environments are highly dynamic and responsive to stimuli, a single environment can represent at best a temporary snapshot of the conditions. Conductance of simulations with several sets of resource pools (e.g., representing temporal variations in exudation profile) can add insights on their effect on trophic interactions and community dynamics. In parallel, confirming predictions made in various environments will support an iterative process that will strengthen the predictive power of the framework and improve its accuracy as a tool for generating testable hypotheses. Similarly, complementing the genomics-based approaches done here with additional layers of 'omics information (mainly transcriptomics and metabolomics) can further constrain the solution space, deflate the number of potential metabolic routes and yield more accurate predictions of GSMMs' performances (*Zampieri et al., 2023*).

To summarize, we have constructed a framework enabling the analysis of metabolic interactions among microbes, as well as between microbes and their hosts, in their natural environment. Where recent studies begin to apply GSMM reconstruction and analysis starting from MAGs (*Zampieri et al., 2023*; *Zorrilla et al., 2021*), this work applies the MAGs to GSMMs approach to conduct large-scale CBM analysis over high-quality MAGs derived from a native rhizosphere and explore the complex network of interactions in light of the functioning of the respective agroecosystem. The application of this framework to the apple rhizobiome yielded a wealth of preliminary knowledge about the metabolic interactions occurring within it, including novel information on putative functions performed by bacteria in healthy vs replant-diseased soil systems, and potential metabolic routes to control these functions. Overall, this framework aims to advance efforts seeking to unravel the intricate world of microbial interactions in complex environments including the plant rhizosphere. The framework is provided as a three stage-detailed pipeline in GitHub, copy archived at *FreilichLab, 2023a*.

## Methods
### Recovery of MAGs from metagenomics data constructed for apple rhizosphere microbiomes

High coverage shotgun metagenomics sequence data were obtained from microbial DNA extracted from the rhizosphere of apple rootstocks cultivated in soil from a replant-diseased orchard (*Berihu et al., 2023*). The experimental design included sampling of six different soil/apple rootstock treatments with five biological replicates each, as described in *Somera et al., 2021*. Two different apple rootstocks (M26, ARD susceptible; G210, ARD tolerant) were grown in three different treatments: (1) orchard soil amended with *Brassica napus* seed meal, (2) orchard soil amended with *Brassica juncea/Sinapis alba* seed meal (BjSa), and (3) no-treatment control soil (NTC) (see *Figure 2—source data 2*). Microbial DNA was extracted from rhizosphere soil and metagenomics data were assembled as described in *Berihu et al., 2023*. In each assembly contigs were binned to recover MAGs using MetaWRAP pipeline (v1.3.1), which utilizes several independent binners (*Uritskiy et al., 2018*). The MAGs recovered by the different binners were collectively processed with the Bin_refinement module of metaWRAP, producing an output of a refined bin collection. A count table was constructed by mapping raw reads data of each assembly (e.g., BjSa; G210) to the bins, using BWA-MEM (Burrows-Wheeler Aligner - Maximum Exact Match) mapping software (version 0.7.17) with default parameters.

DA of the reads associated with the respective bins in each assembly across the respective replicates was determined using the edgeR function implemented in R, requiring FDR adapted p-value <0.05. Read mapping information is shown in *Figure 2—source data 2*. Based on DA, MAGs were classified either as associated with healthy soil (H; BjSa DA), sick soil (S; NTC DA), or not-associated with either soil type (NA; not DA at any treatment site).

Gene calling and annotation were performed with the Annotate_bins module of MetaWRAP. Pathway completeness was determined with KEGG Decoder v 1.0.8.2 (*Graham et al., 2018*) based on the KO annotations extracted from Annotate_bins assignments. The quality of the genomes was determined with CheckM (*Parks et al., 2015*). For phylogenomic analyses and taxonomic classification of each bacterial and archaeal genome, we searched for and aligned 120 bacterial marker genes of the MAGs using the identity and align commands of GTDB-Tk v1.5.0 (*Chaumeil et al., 2019*). MAGs were de-replicated using dRep v2.3.2 (*Olm et al., 2017*) using the default settings and MAGs from the six assemblies were clustered into a single non-redundant set. Phylogenomic trees were rooted by randomly selecting a genome from the sister lineage to the genus as determined from the topology of the bacterial and archaeal GTDB R06-RS202 reference trees. Closely related GTDB taxa identified with the 'classify_wf' workflow were filtered using the taxa-filter option during the alignment step. Overall, a set of 395 high-quality genomes (≥90% completeness,<5% contamination) was used for downstream analyses.

## GSMM reconstruction, analysis, and characterization of the MAG collection

GSMMs were constructed for each of the 395 MAGs using CarveMe v 1.5.1 (*Machado et al., 2018,*) a python-based tool for GSMM reconstruction. Installation and usage of CarveMe were done as suggested in the original CarveMe webpage (https://carveme.readthedocs.io/en/latest/). The solver used for GSMM reconstruction is Cplex (v. 12.8.0.0). All GSMMs were drafted without gap filling as it might mask metabolic co-dependencies (*Opatovsky et al., 2018*). Stoichiometric consistency of all GSMMs was systematically assessed via the standardized MEMOTE test suite, a tool for GSMM quality and completion assessment (*Lieven et al., 2020*). GSMMs not stoichiometrically balanced were filtered out, as they might produce infeasible simulation results.

Analyses and simulations of GSMMs, as well as retrieval of model attributes (reactions, metabolites, exchanges, etc.), were conducted via the vast array of methods found in COBRApy (*Ebrahim et al., 2013*), a python coding language package for analyzing constraint-based reconstructions. For each GSMM, initial growth simulations took place in three different model-specific environments: rich medium, poor medium, and poor medium + exudates. The rich medium was composed of all the exchange reactions (i.e., exchange reaction for specific compounds) a model holds, gathered by the 'exchanges' attribute found in each model. The poor medium was composed of the minimal set of compounds required for a specific GSMM/species to grow at a fixed rate. This set of compounds was identified using the minimal_medium module from COBRApy (minimize components = True, growth rate of 0.1 biomass increase hour$^{-1}$). Poor medium + exudates was defined as the poor medium with the addition of an array of apple-root exudates. These compounds were retrieved from two metabolomics studies characterizing the exudates of apple rootstocks grown in Lane Mountain Sand (Valley, WA) (*Leisso et al., 2017*; *Leisso et al., 2018*). Exudate compounds were aligned with the BIGG database (*King et al., 2016*) in order to format them for use in COBRApy. For each media type, GSMM growth rates were calculated by solving each model using the summary method in COBRApy, which utilizes Flux Balance Analysis (FBA) for maximizing biomass.

## Construction of a common root environment medium and application of the MCSM

In order to simulate the dynamics of the rhizosphere community in the root environment, a fourth growth medium representing a natural-like environment was defined and termed the 'rhizosphere environment'. The rhizosphere environment was composed of two arrays of compounds: (1) the exudates (as described above) and (2) inorganic compounds essential for sustaining bacterial growth. This array was determined according to the minimal set of compounds identified for each GSMM (also described above). Rhizosphere environment components are provided in *Figure 4—source data*

*1*. Both sets of compounds were then consolidated into one array in which further simulations were conducted.

The MCSM (which is the first module out of three comprising this computational framework) simulates the growth of a microbial community by iteratively growing the GSMMs in the community and adding compounds 'secreted' by the growing species to the simulation environment (i.e., the medium), thus enriching the medium/environment and supporting further growth. Unlike FBA, which is used for gathering an arbitrary solution regarding non-optimized fluxes, the MCSM uses FVA to determine exchange fluxes (*Mahadevan and Schilling, 2003*). FVA gathers the full range of exchange fluxes (both secretion and uptake) that satisfy the objective function of a GSMM (i.e., biomass increase). The FVA fraction of optimum was set to 0.9 (sustaining the objective function at 90% optimality, allowing a less restricted secretion profile). Secretion compounds added to the updated medium in each iteration were set to be given in optimal fluxes in next iteration, to ensure a metabolic effect based on the presence of specific metabolites in the environment, rather than their quantity. Flux boundaries of updated medium components were set to 1000 mmol/gDW hour (for a specific exchange compound; gDW, gram Dry Weight).

MCSM was initially simulated in the rhizosphere environment medium. After each growth iteration, GSMM growth values and the compounds secreted by the species growing were collected, where the latter were added to the medium for the next iteration as described above. Growth iterations continued until no new compounds were secreted and no additional GSMMs had grown. After the third iteration, a set of organic phosphorous compounds (containing both carbon and phosphorous) was added to the environment. These compounds were gathered from the pool of model-specific minimal compounds selected for use in the poor medium. Information regarding the chemical formula of these compounds was gathered with the formula attribute of each compound object in COBRApy. Along MCSM iterations, secreted metabolites were classified into biochemical categories based on BRITE annotations (*Aoki-Kinoshita and Kanehisa, 2007*) or, in the absence of classification, manually. MCSM was further applied to inspect the framework's ability to tracing cellulose degradation using cellulose medium (*Figure 4—source data 3*). The tutorial for the MCSM stage of the framework workflow is found in GitHub, along with the GSMMs and media files. Instructions for conducting the analysis are in the README.md file.

## Construction of the exchange network and its untangling for screening sub-network motifs

For each GSMM, a directed bipartite network representing all potential metabolic exchanges occurring within the rhizosphere community was constructed based on uptake and secretion data derived from MCSM iterations. Edges in the network were connected between GSMM nodes and metabolite nodes, with edge directionality indicating either secretion or uptake of a metabolite by a specific GSMM. The network was constructed with the networkx package, a python language package for the exploration and analysis of networks and network algorithms. Network-specific topography was obtained using the Kamada-kawai layout (*Hagberg et al., 2008*). Information regarding the degree and connectivity of the different node types was acquired from the graph object (G). Code for the network construction module in the framework is found at the project's GitHub page under the name NETWORK.py.

Untangling the exchange network into individual paths (i.e., sub-network motifs) was done using the all_shortest_paths function in networkx (*Hagberg et al., 2008*), applied over the exchanges network. Briefly, the algorithm screens for all possible shortest paths within the network, specifically screening for paths starting with an exudate node and ending secreted metabolite nodes (secreted by bacterial species but not consumed). This algorithm yielded two types of paths: (1) PM paths in which node positions one, two, and three represented exudate, microbe, and secreted metabolite, respectively, and (2) PMM paths. PMM paths (length of five nodes) were constructed based on PM paths (length of three nodes), in which positions four and five displayed unique (i.e., not found in PM paths) microbe and metabolite nodes, respectively. Code for the network untangling module in the framework is found at the project's GitHub page under the name PATHS.py.

## Associating PM and PMM sub-network motifs features with soil health

Sub-network motifs (PM and PMM) were functionally classified as associated with healthy soil (H), sick soil (S), or not-associated with either soil type (NA) based on differential abundance of the

corresponding MAGs (PM) or MAGs combination (PMM) in the sub-network. Next, the GSMMs in both PM and PMM sub-networks were characterized according functional classification. For PMMs, the distribution of counts of classified sub-networks, at the different positions, was compared using the ANOVA test, followed by a Tukey test to significantly distinguish the groups. GSMM classifications were further projected on uptake and secreted metabolites in the pathway motifs. For simplification, the analysis focused only on PM paths because PMM paths incorporate PM paths, and exchanges within a PMM path do not directly reflect the effect of an exudate on the secreted end-product (but over the intermediate compound). On that account, start/end metabolites in PM paths were associated with H/S/NA paths based on one-sided hypergeometric test, comparing the frequency of each compound in a functionally characterized path type (either H or S) vs its frequency in NA classified paths and the reciprocal dataset.

## Acknowledgements

We would like to express our gratitude to Asaf Sadeh and Roni Gafni for their advice as part of the Neve Yaar Stats and R support forum, and to Ariana Basile for her advice on conducing quality control tests over the genome-scale metabolic models. We would also like to thank Maya Bar Yehuda for graphical design.

## Additional information

### Funding

| Funder | Grant reference number | Author |
|---|---|---|
| United States-Israel Binational Agricultural Research and Development Fund | US-5390-21 | Shiri Freilich |

The funders had no role in study design, data collection, and interpretation, or the decision to submit the work for publication.

### Author contributions

Alon Avraham Ginatt, Conceptualization, Formal analysis, Investigation, Visualization, Methodology, Writing - original draft, Writing - review and editing; Maria Berihu, Data curation, Formal analysis, Investigation, Visualization, Methodology, Writing - original draft; Einam Castel, Formal analysis, Validation, Investigation; Shlomit Medina, Data curation, Methodology; Gon Carmi, Adi Faigenboim-Doron, Resources, Software, Validation; Itai Sharon, Validation; Ofir Tal, Resources, Validation, Methodology; Samir Droby, Hanan Eizenberg, Supervision, Validation; Tracey Somera, Resources, Supervision, Validation, Investigation; Mark Mazzola, Supervision, Validation, Investigation; Shiri Freilich, Conceptualization, Supervision, Validation, Writing - original draft, Writing - review and editing

### Author ORCIDs

Alon Avraham Ginatt ⓘ https://orcid.org/0009-0004-7817-354X
Shlomit Medina ⓘ http://orcid.org/0000-0001-9239-0576
Gon Carmi ⓘ https://orcid.org/0000-0002-1031-2935
Itai Sharon ⓘ http://orcid.org/0000-0003-0705-2316
Ofir Tal ⓘ http://orcid.org/0009-0003-6997-7418
Tracey Somera ⓘ http://orcid.org/0000-0003-4369-8472
Shiri Freilich ⓘ http://orcid.org/0000-0002-5173-9695

Reviewer #1 (Public review): https://doi.org/10.7554/eLife.94558.3.sa1
Reviewer #3 (Public review): https://doi.org/10.7554/eLife.94558.3.sa2
Author response https://doi.org/10.7554/eLife.94558.3.sa3

# Additional files

## Supplementary files
• MDAR checklist

## Data availability

A reproducible workflow for central computational analyses was deposited in GitHub, copy archived at *FreilichLab, 2023a* including all GSMMs. Raw metagenome sequences used in this study have been deposited in NCBI under BioProject accession number PRJNA779554. MAGs FASTA sequences generated and used in this work are located in GitHub, copy archived at *FreilichLab, 2023b*.

The following previously published dataset was used:

| Author(s) | Year | Dataset title | Dataset URL | Database and Identifier |
|---|---|---|---|---|
| Berihu M, Somera TS, Malik A, Medina S, Piombo E, Tal O, Cohen M, Ginatt A, Ofek-Lalzar M, Doron-Faigenboim A, Mazzola M | 2023 | Metagenome analysis on the apple rhizosphere microbiome | https://www.ncbi.nlm.nih.gov/bioproject/PRJNA779554 | NCBI BioProject, PRJNA779554 |

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
