## [Editor Report · eLife assessment]

By developing a framework to integrate metagenomic and metabolomic data with genome-scale metabolic models, this study establishes a toolkit to investigate trophic interactions between microbiota members in situ. The authors apply this method to the native rhizosphere bacterial communities of apple rootstocks, producing **solid** evidence and numerous detailed hypotheses on specific trophic exchanges and resource dependencies. The framework represents a **valuable** method to disentangle features of microbial interaction networks and will be of interest to microbiome scientists as well as plant and computational biologists.

---

## [Referee Report · Reviewer #1 (Public review)]

The work by Ginatt et al. uses genome-scale metabolic modeling to identify and characterize trophic interactions between rhizosphere-associated bacteria. Beyond identifying microbial species associated with specific host and soil traits (e.g., disease tolerance), a detailed understanding of the interactions underlying these associations is necessary for developing targeted microbiome-centered interventions for plant health. It has nonetheless remained challenging to define the roles of specific organisms and metabolic species in natural rhizobiomes. Here, the authors combine microbial compositional data obtained through metagenomic sequencing with a new collection of genome-scale models to predict interactions in the native rhizosphere communities of apple rootstocks. To do this, they have established processes to integrate these sources of data and model specific trophic exchanges, which they use to obtain testable hypotheses for targeted modulation of microbiota members in situ.

The authors carry out a careful model curation process based on metagenomic sequencing data and existing model generation tools, which, together with basing the in silico medium composition on known root exudates, strengthens their predictions of interaction network features. Moreover, its reliance on genome-scale models provides a broader basis for linking sequence-based information to predictions of function on a multispecies level beyond rhizosphere microbiomes.

Having generated a set of predicted trophic interactions, the authors carried out a detailed analysis linking features of these interactions to organism taxonomy and broader ecosystem properties. Intriguingly, the organisms predicted to grow in the first iteration of their framework (i.e., on only root exudates) broadly correspond to taxonomic groups experimentally shown to benefit from these compounds. Additionally, the simulations predicted some patterns of vitamin and amino acid secretion that are known to form the basis for interactions in the rhizosphere. Together, these outcomes underscore the applicability of this method to help disentangle trophic interaction networks in complex microbiomes.

The methodology described in this paper represents a useful and promising framework to better understand the complexity of microbial interaction networks in situ. In particular, the authors' simulation of trophic interactions based on cellulose degradation have generated predictions of interactions that can more readily be validated. While a more complete analysis of the method's sensitivity to environmental composition is still needed to fully interpret its conclusions - particularly those predicting the inability of many of the in silico organisms to produce biomass - it represents a valuable addition to the growing toolkit of computational and experimental methods for generating educated hypotheses on complex trophic networks.

---

## [Referee Report · Reviewer #3 (Public review)]

Summary:

This study presents a solid framework for the metabolic modeling of microbial species and resources in the rhizosphere environment. It is an ambitious effort to tackle the huge complexity of the rhizosphere and reveal the plant-microbiota interactions therein. Considering previously published data by Berihu et al., going through a series of steps, the framework then finds associations between an apple tree disease state and both microbes and metabolites. The framework is well explained and motivated. I think that further work should be done to validate the method, both using synthetic data, with a known ground truth and following up on key findings experimentally.

Strengths:

- The manuscript is well written with a good balance between detail and readability. The framework steps are well motivated and explained.

- The authors faithfully acknowledge the limitations of their approach and do not try to "over-sell" their conclusions.

- The presented framework has potential for significant discovery if the hypotheses generated are followed up with experimental validation.

Weaknesses:

- It would be better for the framework to be validated on synthetic data.

Justification of claims and conclusions:

The claims and conclusions are sufficiently well justified since the limitations of this approach are acknowledged by the authors.

---

## [Author Response]

The following is the authors’ response to the original reviews.

**Public Reviews:**

**Reviewer #1 (Public review):**
…the degree to which the predictions can vary according to environmental composition remains difficult to quantify, and the work does not address the sensitivity of the modeling predictions beyond a simulated medium containing 33 root exudates. I find this especially important given that relatively few (84 of 243) species were predicted to grow even after cross-feeding, suggesting that a richer medium could lead to different interaction network structures. While the authors do state the importance of environmental composition and have carefully designed an in silico medium, I believe that simulating a broader set of resource pools would add necessary insight into both the predictive power of the models themselves and trophic interactions in the rhizosphere more generally.

The original analyses were indeed focused on a single well-defined environment supporting the growth of only a subset of the species. We have added a paragraph to the discussion section dealing with the potential limitations of this approach.

On line 289 we write:

"Overall, the successive iterations connected 84 out of 243 native members of the apple rhizosphere GSMM community via trophic exchanges. The inability of the remaining bacteria to grow, despite being part of the native root microbiome, possibly reflects the selectiveness of the root environment, which fully supports the nutritional demands of only part of the soil species, whereas specific compounds that might be essential to other species are less abundant1. It is important to note that the specific exudate profile used here represent a snapshot of the root metabolome as root secretion-profiles are highly dynamic, reflecting both environmental and plant developmental conditions. A possible complementary explanation to the observed selective growth might be the partiality of our simulation platform, which examined only plant-bacteria and bacteria-bacteria interactions while ignoring other critical components of the rhizosphere system such as fungi, archaea, protists and mesofauna, as well as less abundant bacterial species, components all known to metabolically interact2. Finally, the MAG collection, while relatively substantial, represents only part of the microbial community. Accordingly, the iterative growth simulations represent a subset of the overall hierarchical-trophic exchanges in the root environment, necessarily reflecting the partiality of the dataset."

In addition, we have tried to better explain the advantages of a limited/defined medium to such an analysis. On Line 231 we add:

"By avoiding the inclusion of non-exudate organic metabolites, the true-to-source rhizosphere environment was designed to reveal the hierarchical directionality of the trophic exchanges in soil, as rich media often mask various trophic interactions taking place in native communities3"

More generally, beyond the above justification of our specific medium selection, we agree that simulating a broader set of resource pools would contribute to a more comprehensive understanding of the trophic interactions. Therefore, we conducted the analysis in an additional environment, in which cellulose was used as an input. We were able to follow its well-documented degradation via multiple steps, conducted by different community members, to serve as a benchmark to our suggested framework.

On line 357 we add:

"To validate the ability of MCSM to capture trophic dependencies and succession, we further tested whether it can trace the well-documented example of cellulose degradation - a multi-step process conducted by several bacterial strains that go through the conversion of cellulose and its oligosaccharide derivatives into ethanol, acetate and glucose, which are all eventually oxidized to CO24. Here, the simulation followed the trophic interactions in an environment provided with cellulose oligosaccharides (4 and 6 glucose units) on the 1st iteration (Supp. Table 3). The formed trophic successions detected along iterations captured the reported multi-step process (Supp.Fig.7)."

Finally, we have included additional text regarding the challenge of defining our simulation environment in the Discussion section.

On line 532 we add:

"In the current study, the root environment was represented by a single pool of resources (metabolites). As genuine root environments are highly dynamic and responsive to stimuli, a single environment can represent, at best, a temporary snapshot of the conditions. Conductance of simulations with several sets of resource pools (e.g., representing temporal variations in exudation profile) can add insights regarding their effect on trophic interactions and community dynamics. In parallel, confirming predictions made in various environments will support an iterative process that will strengthen the predictive power of the framework and improve its accuracy as a tool for generating testable hypotheses. Similarly, complementing the genomicsbased approaches used here with additional layers of 'omics information (mainly transcriptomics & metabolomics) can further constrain the solution space, deflate the number of potential metabolic routes and yield more accurate predictions of GSMMs' performances5."

And we add in Line 520:

"For these reasons, among others, the framework presented here is not intended to be used as a stand-alone tool for determining microbial function. The framework presented is designed to be used as a platform to generate educated hypotheses regarding bacterial function in a specific environment in conjunction with actual carbon substrates available in the particular ecosystem under study. The hypotheses generated provide a starting point for experimental testing required to gain actual, targeted and feasible applicable insights6,7. While recognizing its limitations, this framework is in fact highly versatile and can be used for the characterization of a variety of microbial communities and environments. Given a set of MAGs derived from a specific environment and environmental metabolomics data, this computational framework provides a generic simulation platform for a wide and diverse range of future applications."

**Reviewer #2 (Public review):**
There are two main drawback approaches like the one described here, both related only partially to the authors' work yet with great impact in the presented framework. First, the usage of automatic GSMM reconstruction requires great caution. It is indicative of how the semicurated AGORA models are still considered reconstructions and expect the user to parameterize those in a model. In this study, CarveMe was used. CarveMe is a well-known tool with several pros [1]. Yet, several challenges need to be considered when using it [2]. For example, the biomass function used might lead to an overestimation of auxotrophies. Also, as its authors admit in their reply paper, CarveMe does gap fill in a way [3]; models are constructed to ensure no gaps and also secure a minimum growth. However, curation of such a high number of GSMMs is probably not an option. Further, even if FVA is way more useful than FBA for the authors' aim, it does not yet ensure that when a species secretes one compound (let's say metabolite A), the same flux vector, i.e. the same metabolic functioning profile, secretes another compound (metabolite B) at the same time, even if the FVA solution suggests that metabolite B could be secreted in general.

We thank Reviewer #2 for highlighting this key limitation of our analysis. Below and in the 'recommendations to authors' section we address these concerns.

Concerning the first point raised (models' accuracy) we have now clearly acknowledged in the text the limitations of using an automated GSMM reconstruction tool such as CarveMe. More generally, the framework applied here was built in order to meet the challenges of analyzing highthroughput data while acknowledging the inherent potential of introducing inaccuracies. Pros & cons are now discussed.

On line 507 we write:

"Moreover, the use of an automatic GSMM reconstruction tool (CarveMe8), though increasingly used for depicting phenotypic landscapes, is typically less accurate than manual curation of metabolic models9. This approach typically neglects specialized functions involving secondary metabolism10 and introduces additional biases such as the overestimation of auxotrophies11,12. Nevertheless, manual curation is practically non-realistic for hundreds of MAGs, an expected outcome considering the volume of nowadays sequencing projects. As the primary motivation of this framework is the development of a tool capable of transforming high-throughput, low-cost genomic information into testable predictions, the use of automatic metabolic network reconstruction tools was favored, despite their inherent limitations, in pursuit of addressing the necessity of pipelines systematically analyzing metagenomics data."

Regarding using FVA solutions, indeed such solutions return all potential metabolic fluxes in GSMMs (ranges of all fluxes satisfying the objective function, which by default is set to biomass increase) in a given environment. However, as indicated by the reviewer, predicted fluxes do not necessarily co-occur (i.e., when a metabolite is secreted another metabolite is not necessarily secreted too), yet, they provide the full set of potential solutions (unlike the single solution provided by FBA). A possible strategy to reduce inflated predictions provided by FVA and further constrain the solution space (reduce the set of metabolic fluxes) can be the incorporation of additional `omics data layers, as for example was done in the work of Zampieri et al5. Such approach could allow for instance limiting active reactions (blocking fluxes) from the network reconstructions if not coming to play *in situ*, and therefore impose further constraints and narrow the solution space. We now refer in the text to this limitation and to potential routes to overcome it.

On line 541 we now write:

Similarly, complementing the genomics-based approaches done here with additional layers of 'omics information (mainly transcriptomics & metabolomics) can further constrain the solution space, deflate the number of potential metabolic routes and yield more accurate predictions of GSMMs' performances5.

**Reviewer #3 (Public review):**
When presenting a computational framework, best practices include running it on artificial (synthetic) data where the ground truth is known and therefore the precision and accuracy of the method may be assessed. This is not an optional step, the same way that positive/negative controls in lab experiments are not optional. Without this validation step, the manuscript is severely limited. The authors should ask themselves: what have we done to convince the reader that the framework actually works, at least on our minimal synthetic data?

Thank you for this suggestion. To validate the ability of MCSM to capture trophic succession, we conducted an additional analysis testing whether it can track the well documented example of cellulose degradation - a multi-step process conducted by several bacterial strains. This example has been included in the manuscript to serve as a case study (i.e. positive control) for metabolic interactions occurring within the bacterial community (Supp. Fig. 7).

On line 357 we add:

"To validate the ability of MCSM to capture trophic dependencies and succession, we further tested whether it can track the well-documented example of cellulose degradation - a multi-step process conducted by several bacterial strains that go through the conversion of cellulose and its oligosaccharide derivatives into ethanol, acetate and glucose, which are all eventually oxidized to CO24. Here, the simulation followed the trophic interactions in an environment provided with cellulose oligosaccharides (4 and 6 glucose units) on the 1st iteration (Supp. Table 3). The formed trophic successions detected along iterations captured the reported multi-step process (Supp. Fig.7)."

"Supplementary Figure 7. Application of MCSM over the process of cellulose decomposition as described by Kato et al4. 5-partite network exhibiting the uptake of cellulose oligomers (4 and 6 units of connected D-glucose) by primary decomposers, through secretion of intermediate compounds and their metabolization by secondary decomposers to CO2. Distribution of phyla of primary and secondary decomposers is denoted by pie charts. Though MAGs were not constructed for the original species as in Kato et al., among the primary consumers, species corresponding to the Acidobacteria (Acidobacteriales)13, Actinobacteria14, Bacteriodetes15, Proteobacteria (Xanthomonadales)16 and Verrucobacteria17 groups are found to be capable of degrading cellulose compounds via enzymatic mechanisms."

More generally, beyond the above addition, the relevance of the framework to the analysis of the data is discussed throughout the analysis (in the original version of the manuscript). We have scrutinized each of our observations in light of current available information and provided a corroborating evidence as well as a few discrepancies for multiple steps in the analysis. Examples include the following discussions:

On line 312, we discuss the biological relevance of taxonomic classes classified as primary versus secondary degraders

"As in the full GSMM data set (Community bar, Fig. 3C), most of the species which grew in the 1st iteration belonged to the phyla *Acidobacteriota*, *Proteobacteria,* and *Bacteroidota*. This result concurred with findings from the work of Zhalnina et al, which reported that bacteria assigned to these phyla are the primary beneficiaries of root exudates18. Species from three out of the 17 phyla that did not grow in the first iteration - *Elusimicrobiota, Chlamydiota*, and *Fibrobacterota,* did grow on the 2nd iteration (Fig. 3C). Members of these phyla are known for their specialized metabolic dependencies. Such is the case for example with members of the *Elusimicrobiota* phylum, which include mostly uncultured species whose nutritional preferences are likely to be selective19.

At the order level, bacteria classified as *Sphingomonadales* (class *Alphaproteobacteria*), a group known to include typical inhabitants of the root environment20, grew in the initial Root environment. In comparison, other root-inhabiting groups including the orders *Rhizobiales* and _Burkholderiales_20, did not grow in the first iteration. *Rhizobiales* and *Burkholderiales* did, however, grow in the second and third iterations, respectively, indicating that in the simulations, the growth of these groups was dependent on exchange metabolites secreted by other community members (Supp. Fig. 4)."

On line 331, we provide support to the classification of specific metabolites as exchange molecules

"Overall, 158 organic compounds were secreted throughout the MCSM simulation (from which 12 compounds overlapped with the original exudate medium). These compounds varied in their distribution and were mapped into 12 biochemical categories (Fig. 3D). Whereas plant secretions are a source of various organic compounds, microbial secretions provide a source of multiple vitamins and co-factors not secreted by the plant. Microbial-secreted compounds included siderophores (staphyloferrin, salmochelin, pyoverdine, and enterochelin), vitamins (pyridoxine, pantothenate, and thiamin), and coenzymes (coenzyme A, flavin adenine dinucleotide, and flavin mononucleotide) – all known to be exchange compounds in microbial communities21,22. In addition, microbial secretions included 11 amino acids (arginine, lysine, threonine, alanine, serine, phenylalanine, tyrosine, leucine, glutamate, isoleucine, and methionine), also known as a common exchange currency in microbial communities23. Some microbial-secreted compounds, such as phenols and alkaloids, were reported to be produced by plants as secondary metabolites24,25. Additional information regarding mean uptake and secretion degrees of compounds classified to biochemical groups is found in Supp. Fig. 5."

On line 432, we provide corroborative support to the classification of exudates as associated with beneficial/non beneficial root communities

"Notably, the S-classified root exudates included compounds reported to support dysbiosis and ARD progression. For example, the S-classified compounds gallic acid and caffeic acid (3,4-dihidroxy-trans-cinnamate) are phenylpropanoids – phenylalanine intermediate phenolic compounds secreted from plant roots following exposure to replant pathogens26. Though secretion of these compounds is considered a defense response, it is hypothesized that high levels of phenolic compounds can have autotoxic effects, potentially exacerbating ARD. Additionally, it was shown that genes associated with the production of caffeic acid were upregulated in ARD-infected apple roots, relative to those grown in γ-irradiated ARD soil27,28, and that root and soil extracts from replant-diseased trees inhibited apple seedling growth and resulted in increased seedling root production of caffeic acid29."

On line 446, we provide a supporting evidence to the classification of secreted compounds as associated with beneficial/non beneficial root communities

"Several secreted compounds classified as healthy exchanges (H) were reported to be potentially associated with beneficial functions. For instance, the compounds L-Sorbose (EX_srb__L_e) and Phenylacetaladehyde (EX_pacald_e), both over-represented in H paths (Fig. 5C), have been shown to inhibit the growth of fungal pathogens associated with replant disease30,31.

Phenylacetaladehyde has also been reported to have nematicidal qualities32."

On line 453 we discuss the correspondence of specific exudate uptakes and compound secretions via specific subnetwork motifs (PM) and their literature/experimental evidence

"Combining both exudate uptake data and metabolite secretion data, the full H-classified PM path 4-Hydroxybenzoate; GSMM_091; catechol (Fig. 4C; the consumed exudate, the GSMM, and the secreted compound, respectively) provides an exemplary model for how the proposed framework can be used to guide the design of strategies which support specific, advantageous exchanges within the rhizobiome. The root exudate 4-Hydroxybenzoate is metabolized by GSMM_091 (class *Verrucomicrobiae*, order *Pedosphaerales*) to catechol. Catechol is a precursor of a number of catecholamines, a group of compounds which was recently shown to increase apple tolerance to ARD symptoms when added to orchard6,33. This analysis (PM; Fig 4C), leads to formulating the testable prediction that 4-Hydroxybenzoate can serve as a selective enhancer of catecholamine synthesizing bacteria associated with reduced ARD symptoms, and therefore serve as a potential source for indigenously produced beneficial compounds."

Moreover, we perceive our analysis as a strategy for integrating high throughput genomic data into testable predictions allowing narrowing the solution space while acknowledging potential inaccuracies that are inherent to the analysis. We have revised the text in order to clearly acknowledge this limitation.

On line 497 we write:

"The framework we present is currently conceptual."

On line 520 we write:

"For these reasons, among others, the framework presented here is not intended to be used as a stand-alone tool for determining microbial function. The framework presented is designed to be used as a platform to generate educated hypotheses regarding bacterial function in a specific environment in conjunction with actual carbon substrates available in the particular ecosystem under study. The hypotheses generated provide a start point for experimental testing required to gain actual, targeted and feasibly applicable insights6,7."

On line 532 we add:

"In the current study, the root environment was represented by a single pool of resources (metabolites). As genuine root environments are highly dynamic and responsive to stimuli, a single environment can represent, at best, a temporary snapshot of the conditions. Conductance of simulations with several sets of resource pools (e.g., representing temporal variations in exudation profile) can add insights regarding their effect on trophic interactions and community dynamics. In parallel, confirming predictions made in various environments will support an iterative process that will strengthen the predictive power of the framework and improve its accuracy as a tool for generating testable hypotheses. Similarly, complementing the genomicsbased approaches used here with additional layers of 'omics information (mainly transcriptomics & metabolomics) can further constrain the solution space, deflate the number of potential metabolic routes and yield more accurate predictions of GSMMs' performances5."

**Recommendations for the authors:**

**Reviewer #1 (Recommendations for the authors):**
(1) Line 219: "Feasibility" - this term/concept may be difficult to understand for readers unfamiliar with GSMMs. I would recommend either clarifying or rephrasing, perhaps as "simulations confirmed the existence of a feasible solution space for all the 243 models, as well as their capacity to predict growth in the respective environment."

Thanks, done. We have modified this section as suggested (line 221).

(2) Line 244: How does MCSM fit within/build upon existing frameworks that simulate patterns of niche construction and cross-feeding with constraint-based modeling?

This is now addressed. On line 250 we write:

"Unlike tools designed for modelling microbial interactions34,35, MCSM bypasses the need for defining a community objective function as the growth of each species is simulated individually. Trophic interactions are then inferred by the extent to which compounds secreted by bacteria could support the growth of other community members."

(3) Figure 4A: While illustrating the general complexity of the predicted trophic interactions, the density of the network makes it very difficult to interpret specific exchanges. Moreover, the naming conventions of the metabolites make it difficult to understand what they represent. I would recommend either restructuring the graph such that the label of each node is legible, or removing the labels altogether.

Thanks, done. Labels were removed and a zoom-in-window to the exchanges highlighted in Figure 4C were added. Caption was revised to indicate that node colors correspond to differential abundance classification of GSMMs in the different plots (H, S, NA are Healthy, Sick, Not-Associated, respectively).

**Reviewer #2 (Recommendations for the authors):**
CarveMe solves a Mixed Integer Linear Program (MILP) that enforces network connectivity, thus requiring gapless pathways. It's puzzling how to deal with such a great number of GSMMs that is for sure, especially when coming from such an environment as soil and the vast majority of their corresponding MAGs represent most likely novel taxa. One alternative approach for using CarveMe might be to use the rich medium as a medium to gap-fill during the reconstruction. In this case, the gene annotation scores that CarveMe calculates in its initial step, are used to prioritise the reactions selected for gap-filling. This would lead to a new series of challenges but might be a useful comparison with the current GSMMs of the study.

Though indeed CraveMe includes a gap-filling option, here we have purposely avoided the gapfilling option as we aimed to adhere to genomic content of the corresponding genomes and to avoid masking their metabolic dependencies emerging due to their incompleteness. This is noted in the Methods section, which we revised to emphasize the adherence to the genomic content of the models:

On line 615 we now write:

"All GSMMs were drafted without gap filling in order to adhere to genomic content and to avoid masking metabolic co-dependencies51"

More generally, we now refer to the limitation of automatic reconstruction in the context of the current analysis. On line 507 we write:

"Moreover, the use of an automatic GSMM reconstruction tool (CarveMe8), though increasingly used for depicting phenotypic landscapes, is typically less accurate than manual curation of metabolic models9. This approach typically neglects specialized functions involving secondary metabolism10 and introduces additional biases such as the overestimation of auxotrophies11,12. Nevertheless, manual curation is practically non-realistic for hundreds of MAGs, an expected outcome considering the volume of nowadays sequencing projects. As the primary motivation of this framework is the development of a tool capable of transforming high-throughput, low-cost genomic information into testable predictions, the use of automatic, semi-curated, metabolic network reconstruction tools was favored, despite their inherent limitations, in pursuit of developing pipelines for the systematic analysis of metagenomics data."

Thermodynamically infeasible loops have been a challenge in constraint-based analysis [1].However, for the case of FBA and FVA time efficient implementations are already available. Therefore, I would suggest using the loopless flag of the cobrapy package when performing FVA.Also, it would be nice to show/discuss how many exchange reactions each GSMM includes and what is the number of those with at least a non-zero minimum or maximum in the FVA using each of the three media.

Done. In Supplementary Figure 4, we added a graphic summary of active FVA ranges for each GSMM in the three different environments (exchange reactions, non-zero flux). Additionally, we analyzed a subset of models and compared their regular FVA results vs loopless FVA results.

On line 217 we write:

"The number of active exchange fluxes in each medium corresponds with the respective growth performances displaying noticably higher number of potentially active fluxes in the rich environment (also when applying loopless FVA) (Supp. Fig. 4). Overall, Simulations confirmed the existence of a feasible solution space for all the 243 models as well as their capacity to predict growth in the respective environemnt (Supp. Data 5)."

"Supplementary Figure 4. FVA performances of GSMMs in different environments (Supp. Fig.3; Supp. Data 5). A. Distribution of potentially active exchange reactions (non-zero minimum FVA flux) in the different environments. Solid line inside each violin indicates the interquartile range (IQR). White point in IQR indicates the median value. Whiskers extending from the IQR indicate the range within 1.5 times the IQR from the quartiles. Violin width at a given value represents the density of data points at that value. B. Loopless FVA scores compared to regular FVA for models in the 3 different environments. Bars indicate the count of active fluxes (nonzero minimum FVA flux). Only a subset of models was used for this analysis."

This brings us to the main challenge of your framework in my opinion: FVA returns the minimum and the maximum a flux may get. However, it does not ensure that when a metabolite is being secreted, another does the same too. That could lead to an overrepresentation of secreted metabolites after each iteration. To my understanding, unbiased methods focusing on metabolite exchanges would be a much better alternative for such questions. Unbiased constraint-based methods are known for requiring essential computational requirements, yet when focusing on specific parts of the models, recent implementations support them. A great showcase of such techniques is presented in [2].

Indeed, FVA solutions return all *potential* metabolic fluxes in GSMMs (ranges of all fluxes satisfying the objective function, which by default is set to biomass increase) but they do not ensure that all fluxes actually co-occur (i.e., when a metabolite is secreted necessarily another metabolite is secreted too). However, though FVA solutions do not necessarily ensure cooccurrence regarding secretion and uptake, they provide a broader metabolic picture (the full set of potential solutions), unlike the arbitrary single solution provided by FBA, which is limited in providing information about potential secretions and uptakes in a specific environment. Here, we tried to elucidate the connection between a specific environment (root exudates) and the growth and metabolic capabilities of native bacteria. To the best of our understanding, unbiased approaches (such as the one displayed in Wedmark et al.36) are not environment dependent but rather calculate all possible metabolic elements and routes within a metabolic network. Therefore, using FVA is well adapted to explore environment-dependent growth. The sensitivity of FVA predicted active fluxes to the environments is now also implied by Sup. Fig. 3B demonstrating the number of potential active fluxes is proportional to growth performances. In addition, inquiring all possible metabolic routes across a large dataset of hundreds of MAGS, is central to the current analysis, thus the easy implementation of FVA further justifies its use in the current study.

An alternative strategy to reduce inflated FVA predictions and further constrain the solution space of predicted active fluxes can be the incorporation of additional layers of `omics data, as for example was done in the work of Zampieri et al5. Such approach could allow for instance removing reactions from the network reconstructions if not coming to play *in situ*, and therefore impose further constraints and narrow down the solution space. Currently, the complexity of the soil community might impede or at least constrain a high coverage recovery of transcriptomic data, though future works utilizing additional layers of `omics data are expected to significantly reduce the number of potential solutions and thus improve the accuracy of GEMs predictions.

This is now discussed in the text. In line 541 we write:

"Similarly, complementing the genomic-based approaches done here, with additional layers of 'omics information (mainly transcriptomics & metabolomics) can further constrain the solution space, deflate the number of potential metabolic routes and yield more accurate predictions of GSMMs' performances5."

In case it was the first version of CheckM used, the authors could consider repeating this check with CheckM2. As they state in line 293, Archaea may play an essential role in the community. Yet, among the high-quality MAGs only one corresponded to Archaea. However, that is quite possible to be the case because CheckM underestimates the completeness of archaeal genomes. If CheckM2 suggests that archaeal MAGs could be used, these would probably benefit a lot for the aim of the study.

The analysis was conducted with the first version of CheckM to assess MAGs quality. In future analyses we will use CheckM2. However, also before MAG recovery, we already know from the work of Beirhu et al., that Archaea species have a very low representation in the metagenomics data used here (Berihu et al., Additional data 2. Supp. fig. 4; "others" group)6, with less than 0.5% of the contigs mapped to archaeal genomes. The overall taxonomic distribution of the high-quality MAGs was compared to the distribution inferred from the non-binned data (contigs) and amplicon sequencing and the three different data sets are very similar (Fig. 2).

On line 130 we write:

"Overall, the taxonomic distribution of the MAG collection corresponded with the profile reported for the same samples using alternative taxonomic classification approaches such as 16S rRNA amplicon sequencing and gene-based taxonomic annotations of the non-binned shotgun contigs

(Fig. 2B)."

The visualisation of the network in Figure 4A is hard to follow. An alternative could be a 5partite plot having taxa in columns one, three, and five and compounds in the other two. An alternative visualisation is necessary.

The full list of the 5 and 3 partite graphs is provided in supplementary data 10 (also noted in the figure legend now). Figure 4 was revised to improve its visualization. Labels were removed and a zoom in to 5 and 3 partite plots were added (PMM and PM subnetworks, respectively).

Line 509: If I get the point of the authors right, they refer to the "from shotgun data to GEMs" approach. I would suggest skipping this statement. Here is a recent study implementing this: https://doi.org/10.1016/j.crmeth.2022.100383.

Thank you for your comment and reference. The intention behind the phrase in line 509 (in previous version) was to refer to going from metagenomics data to GEMs in soil-rhizosphere microbiome while linking environmental inputs (crop-plants exudates metabolomics data) and the agricultural-related metabolic function of bacteria. This phrase has been modified to clearly make a more modest claim while acknowledging other related studies.

On line 548 we write

"Where recent studies begin to apply GSMM reconstruction and analysis starting from MAGs5,37 , this work applies the MAGs to GSMMs approach to conduct a large-scale CBM analysis over highquality MAGs derived from a native rhizosphere and explore the complex network of interactions in light of the functioning of the respective agro-ecosystem. "

Line 820: Reference format is broken.

Corrected.

In the caption of Figure 4, please add the meaning of H, S, and NA so it is selfexplanatory.

Done. In Figure 4 legend we added:

"Node colors correspond to differential abundance classification of GSMMs in the different plots; H, S, NA are Healthy, Sick, Not-Associated, respectively."

**Reviewer #3 (Recommendations for the authors):**
(1) Figure 4A is unreadable. It is not clear what insight the reader could gain by examining this figure.

Thanks. Figure was revised. Labels were removed and a zoom-in-window to the exchanges highlighted in Figure 4C were added. Caption was revised to indicate that node colors correspond to differential abundance classification of GSMMs in the different plots (H, S, NA are Healthy, Sick, Not-Associated, respectively).

(2) In Figure 5, it is not apparent what the units of "prevalence" are, that is, what is the scale. What does 140 mean? How does that compare to 350?

Thanks. Prevalence in the context of Figure. 5B,C refers to the count of the compounds in each category (significantly affiliated with either healthy or symptomized soils) in sub-network motifs corresponding to this DA classification. We revised the figures (Y axes) and legend to be more specific (B: # of exudates; C: # of secreted compounds).

"B. Bar plot indicating the *number* of exudates significantly associated with H or S-classified PM sub-networks (Hypergeometric test; FDR <= 0.05; green: healthy-H, red: sick-S). C. Bar plots indicate the *number* of secreted compounds in PM sub-networks, which are significantly associated with H-classified (upper, colored green), or S-classified (lower, colored red) (Hypergeometric test; FDR <= 0.05)."

References

(1) Buée, M., de Boer, W., Martin, F., van Overbeek, L. & Jurkevitch, E. The rhizosphere zoo: An overview of plant-associated communities of microorganisms, including phages, bacteria, archaea, and fungi, and of some of their structuring factors. *Plant Soil* 321, 189– 212 (2009).

(2) Bardgett, R. D. & Van Der Putten, W. H. Belowground biodiversity and ecosystem functioning. *Nature* 515, 505–511 (2014).

(3) Opatovsky, I. *et al.* Modeling trophic dependencies and exchanges among insects’ bacterial symbionts in a host-simulated environment. *BMC Genomics* 19, 1–14 (2018).

(4) Kato, S., Haruta, S., Cui, Z. J., Ishii, M. & Igarashi, Y. Stable coexistence of five bacterial strains as a cellulose-degrading community. *Appl. Environ. Microbiol.* 71, 7099–7106 (2005).

(5) Zampieri, G., Campanaro, S., Angione, C. & Treu, L. Metatranscriptomics-guided genomescale metabolic modeling of microbial communities. *Cell Reports Methods* 3, 100383 (2023).

(6) Berihu, M. *et al.* A framework for the targeted recruitment of crop ‑ beneficial soil taxa based on network analysis of metagenomics data. *Microbiome* 1–21 (2023) doi:10.1186/s40168-022-01438-1.

(7) Dhakar, K. *et al.* Modeling-Guided Amendments Lead to Enhanced Biodegradation in Soil. *mSystems* 7, (2022).

(8) Machado, D., Andrejev, S., Tramontano, M. & Patil, K. R. Fast automated reconstruction of genome-scale metabolic models for microbial species and communities. *Nucleic Acids Res.* 46, 7542–7553 (2018).

(9) Henry, C. S. *et al.* High-throughput generation, optimization and analysis of genome-scale metabolic models. *Nat. Biotechnol.* 28, 977–982 (2010).

(10) Freilich, S. *et al.* Competitive and cooperative metabolic interactions in bacterial communities. *Nat. Commun.* 2, (2011).

(11) Price, M. Erroneous predictions of auxotrophies by CarveMe. *Nat. Ecol. Evol.* 7, 194–195 (2023).

(12) Machado, D. & Patil, K. R. Reply to: Erroneous predictions of auxotrophies by CarveMe. *Nat. Ecol. Evol.* 7, 196–197 (2023).

(13) Kulichevskaya, I. S. *et al.* Acidicapsa borealis gen. nov., sp. nov. and Acidicapsa ligni sp. nov., subdivision 1 Acidobacteria from Sphagnum peat and decaying wood. *Int. J. Syst. Evol. Microbiol.* 62, 1512–1520 (2012).

(14) Depart-, M. & Building, L. S. Lignocellulose-degrading actinomycetes. 46, 145–163 (1987).

(15)Thomas, F., Hehemann, J. H., Rebuffet, E., Czjzek, M. & Michel, G. Environmental and gut Bacteroidetes: The food connection. *Front. Microbiol.* 2, 1–16 (2011).

(16) Dow, J. M. & Daniels, M. J. Pathogenicity determinants and global regulation of pathogenicity of Xanthomonas campestris pv. campestris. *Curr. Top. Microbiol. Immunol.* 192, 29–41 (1994).

(17) Bergmann, G. T. *et al.* The under-recognized dominance of Verrucomicrobia in soil bacterial communities. *Soil Biol. Biochem.* 43, 1450–1455 (2011).

(18) Zhalnina, K. *et al.* Dynamic root exudate chemistry and microbial substrate preferences drive patterns in rhizosphere microbial community assembly. *Nat. Microbiol.* 3, 470–480 (2018).

(19) Uzun, M. *et al.* Recovery and genome reconstruction of novel magnetotactic Elusimicrobiota from bog soil. *ISME J.* 1–11 (2022) doi:10.1038/s41396-022-01339-z.

(20) Lei, S. *et al.* Analysis of the community composition and bacterial diversity of the rhizosphere microbiome across different plant taxa. *Microbiologyopen* 8, 1–10 (2019).

(21) Ghosh, S. K., Banerjee, S. & Sengupta, C. Bioassay, characterization and estimation of siderophores from some important antagonistic fungi. *J. Biopestic.* 10, 105–112 (2017).

(22) Lu, X., Heal, K. R., Ingalls, A. E., Doxey, A. C. & Neufeld, J. D. Metagenomic and chemical characterization of soil cobalamin production. *ISME J.* 14, 53–66 (2020).

(23) Mee, M. T., Collins, J. J., Church, G. M. & Wang, H. H. Syntrophic exchange in synthetic microbial communities. *Proc. Natl. Acad. Sci. U. S. A.* 111, (2014).

(24) Justin, K., Edmond, S., Ally, M. & Xin, H. Plant Secondary Metabolites: Biosynthesis, Classification, Function and Pharmacological Properties. *J. Pharm. Pharmacol.* 2, 377–392 (2014).

(25) Yang, W. *et al.* A Genomic Analysis of Bacillus megaterium HT517 Reveals the Genetic Basis of Its Abilities to Promote Growth and Control Disease in Greenhouse Tomato. *Genet. Res. (Camb).* 2022, (2022).

(26) Balbín-Suárez, A. *et al.* Root exposure to apple replant disease soil triggers local defense response and rhizoplane microbiome dysbiosis. *FEMS Microbiol. Ecol.* 97, 1–14 (2021).

(27) Weiß, S., Liu, B., Reckwell, D., Beerhues, L. & Winkelmann, T. Impaired defense reactions in apple replant disease-Affected roots of Malus domestica ‘M26’. *Tree Physiol.* 37, 1672–1685 (2017).

(28) Weiß, S., Bartsch, M. & Winkelmann, T. Transcriptomic analysis of molecular responses in Malus domestica ‘M26’ roots affected by apple replant disease. *Plant Mol. Biol.* 94, 303– 318 (2017).

(29) Sun, N. *et al.* Effects of Organic Acid Root Exudates of Malus hupehensis Rehd. Derived from Soil and Root Leaching Liquor from Orchards with Apple Replant Disease. *Plants* 11, (2022).

(30) Howell, C. R. Seed Treatment with L-Sorbose to Control Damping-Off or Cotton Seedlings by Rhizoctonia solani. *Phytopathology* 68, 1096 (1978).

(31) Zou, C. S., Mo, M. H., Gu, Y. Q., Zhou, J. P. & Zhang, K. Q. Possible contributions of volatile-producing bacteria to soil fungistasis. *Soil Biol. Biochem.* 39, 2371–2379 (2007).

(32) Gomes, V. A. *et al.* Activity of papaya seeds (Carica papaya) against Meloidogyne incognita as a soil biofumigant. *J. Pest Sci. (2004).* 93, 783–792 (2020).

(33) Gao, T. *et al.* Exogenous dopamine and overexpression of the dopamine synthase gene MdTYDC alleviated apple replant disease. *Tree Physiol.* 41, 1524–1541 (2021).

(34) Diener, C., Gibbons, S. M. & Resendis-Antonio, O. MICOM: Metagenome-Scale Modeling To Infer Metabolic Interactions in the Gut Microbiota. *mSystems* 5, (2020).

(35) Dukovski, I. *et al.* A metabolic modeling platform for the computation of microbial ecosystems in time and space (COMETS). *Nat. Protoc.* 16, 5030–5082 (2021).

(36) Katarina Wedmark, Y., Olav Vik, J. & Øyås, O. A hierarchy of metabolite exchanges in metabolic models of microbial species and communities. *bioRxiv* 1–19 (2023).

(37) Zorrilla, F., Buric, F., Patil, K. R. & Zelezniak, A. MetaGEM: Reconstruction of genome scale metabolic models directly from metagenomes. *Nucleic Acids Res.* 49, (2021).